# Local membrane source gathering by p62 body drives autophagosome formation

Xuezhao Feng[1,2,9], Daxiao Sun [3,9] ✉, Yanchang Li[4,9], Jinpei Zhang[1,2,9], Shiyu Liu[1,2], Dachuan Zhang[5], Jingxiang Zheng[5], Qing Xi[1,2], Haisha Liang [5], Wenkang Zhao[5], Ying Li[5], Mengbo Xu[1,2], Jiayu He[1,2], Tong Liu[1,2], Ayshamgul Hasim[2,6], Meisheng Ma[7], Ping Xu [4] ✉ & Na Mi [1,2,8] ✉

Autophagosomes are double-membrane vesicles generated intracellularly to encapsulate substrates for lysosomal degradation during autophagy. Phase separated p62 body plays pivotal roles during autophagosome formation, however, the underlying mechanisms are still not fully understood. Here we describe a spatial membrane gathering mode by which p62 body functions in autophagosome formation. Mass spectrometry-based proteomics reveals significant enrichment of vesicle trafficking components within p62 body. Combining cellular experiments and biochemical reconstitution assays, we confirm the gathering of ATG9 and ATG16L1-positive vesicles around p62 body, especially in *Atg2ab* DKO cells with blocked lipid transfer and vesicle fusion. Interestingly, p62 body also regulates ATG9 and ATG16L vesicle trafficking flux intracellularly. We further determine the lipid contents associated with p62 body via lipidomic profiling. Moreover, with in vitro kinase assay, we uncover the functions of p62 body as a platform to assemble ULK1 complex and invigorate PI3KC3-C1 kinase cascade for PI3P generation. Collectively, our study raises a membrane-based working model for multifaceted p62 body in controlling autophagosome biogenesis, and highlights the interplay between membraneless condensates and membrane vesicles in regulating cellular functions.

Autophagy is an evolutionarily conserved process by which intracellular molecules, damaged organelles or invading pathogens are engulfed into a double-membrane autophagosome and then transported to lysosome for degradation[1]. As autophagy is implicated in many pathophysiological processes such as cellular homeostasis, cancer and neurodegenerative diseases, it is important to understand how autophagy is spatiotemporally regulated[2,3]. Several autophagy-related genes (ATGs) have been identified playing critical roles in autophagosome formation and maturation[4–6]. In mammals, the biogenesis of autophagosome begins with the activation and recruitment of ULK1

[1]State Key Laboratory of Pathogenesis, Prevention and Treatment of Central Asian High Incidence Diseases, Clinical Medical Research Institute, The First Affiliated Hospital of Xinjiang Medical University, Urumqi 830011 Xinjiang, China. [2]Basic Medical College, Xinjiang Medical University, Urumqi 830011 Xinjiang, China. [3]Max Planck Institute of Molecular Cell Biology and Genetics, Pfotenhauerstrasse 108, 01307 Dresden, Germany. [4]State Key Laboratory of Proteomics, National Center for Protein Sciences (Beijing), Research Unit of Proteomics & Research and Development of New Drug of Chinese Academy of Medical Sciences, Beijing Proteome Research Center, Institute of Lifeomics, 102206 Beijing, China. [5]School of Life Sciences, Tsinghua University, 100084 Beijing, China. [6]Department of Pathology, School of Basic Medicine, Xinjiang Medical University, Urumqi 830011 Xinjiang, China. [7]Tongji Medical College of Huazhong University of Science and Technology, Wuhan, Hubei, China. [8]Key Laboratory of High Incidence Disease Research in Xinjiang (Xinjiang Medical University), Ministry of Education, Urumqi 830011 Xinjiang, China. [9]These authors contributed equally: Xuezhao Feng, Daxiao Sun, Yanchang Li, Jinpei Zhang. ✉e-mail: dsun@mpi-cbg.de; xuping@ncpsb.org.cn; mina2a@163.com

complex (composed of ULK1, ULK2, FIP200, ATG13 and ATG101) to a cup-shaped isolation membrane or phagophore. The class III phosphatidylinositol 3-kinase complex I (PI3KC3-C1) (primarily containing VPS34, VPS15, ATG14 and Beclin1) is activated at phagophore membrane to generate phosphatidylinositol 3-phosphate (PI3P) which in turn recruit downstream effectors such as ATG2 and WIPI2. A third complex ATG12–ATG5–ATG16L1 is assembled at nascent phagophore through interaction with WIPI2 and is responsible for the conjugation of LC3 to the membrane lipid phosphatidylethanolamine (PE). These complexes coordinate to initiate autophagosome formation[7,8]. To expand autophagosome, ATG2 (ATG2A and ATG2B) can transfer phospholipids from multiple sources such as endoplasmic reticulum (ER) to feed autophagosome membrane[9–12]. ATG9 (ATG9A and ATG9B), the only integrated transmembrane protein among the core ATGs, not only provides ATG9 vesicles as the seed for phagophore growth, but also acts as phospholipid scramblase to balance the lipid distribution on the autophagosome membrane[13–15]. Apparently, the membrane seeding and expansion are critical prerequisite for autophagosome formation[16,17], however, the detailed mechanisms behind these membrane gathering processes during autophagosome biogenesis are still not fully understood.

When selected autophagic cargos are degraded through autophagy, a group of cargo receptors such as p62/SQSTM1, NBR1, TAX1BP1, NDP52, and OPTN are required to initiate autophagosome formation[7,18–23]. As the first autophagic cargo receptor identified in mammals, p62 interacts with both ubiquitinated proteins and LC3 at phagophore membrane via ubiquitin-associated (UBA) domain and LC3-interaction region (LIR) respectively[20,24]. p62 can also self-oligomerize via its N-terminal PB1 domain[25–27]. The association between p62 and polyubiquitinated proteins drives the formation of p62 body via liquid-liquid phase separation (LLPS), which is important for autophagic cargo segregation and degradation[25,27–31]. The phase separated p62 bodies can serve as a platform for autophagosome biogenesis by tethering LC3-decorated isolation membrane through direct p62-LC3 interaction[32,33]. On the other hand, p62 can interact with FIP200 and thus helps to recruit ULK1 complex or directly binds to ULK1 to initiate autophagosome formation[34,35]. In addition, post-translational modification (for instance, p62 acetylation and phosphorylation) and other cargo receptors also affect the formation of p62 bodies as well as their autophagic clearance[36,37]. Our recent work shows that cytoskeleton dynamics can promote the coalescence of small nanoscale p62 condensates into large micron-scale p62 bodies[38]. Importantly, only large p62 bodies, but not small p62 condensates, can act as a nucleation site for autophagy and be engulfed into mature autophagosome for degradation[38], highlighting the functional importance of the size control of p62 bodies in autophagy. Mutations of p62 affecting its phase separation properties are intimately associated with human diseases[39]. Despite these advances, it remains unknown what constituents partition into p62 bodies as substrates or autophagy machinery and how p62 bodies function in autophagosome formation with proteins and membrane structures cooperatively.

In this study, we systematically profiled the protein composition of p62 bodies with mass spectrometry-based proteomics, and further showed that p62 bodies can start autophagosome generation by gathering ATG9- and ATG16-decorated vesicles as membrane sources and initiator for phagophore nucleation, which is then followed by ATG2A/2B mediated membrane elongation with direct lipid transfer and vesicle fusion. Together with recruited ULK1 and PI3KC3-C1, p62 body serves as a platform for PI3P generation and autophagosome maturation. All together, we revealed a mechanism by which phase separated p62 body cooperates membrane structures and proteins on a membraneless condensate to generate a double membrane autophagosome. Our work fills the gap in previous models and these mechanisms collectively suggest that p62 bodies play multifaceted roles in autophagosome biogenesis.

## Results

### Proteomics profiling reveals significant enrichment of membrane-associated proteins in p62 bodies

To profile the proteins specifically concentrated within the p62 body, we optimized our previous workflow of ubiquitin-induced phase separation of p62/SQSTM1 in vitro[28] to adapt it for mass spectrometry analysis. The recombinant mCherry-p62, Ub8 linear chain and cytosol from HEK293T cells were mixed to form p62 droplet, and the droplet was collected by centrifugation (Fig. 1a and Supplementary Fig. 1a). The mixture (Mix), flow through (FT) and droplet were reduced, alkylated and digested by trypsin in-gel in parallel. The p62 and ubiquitin chains were obviously enriched in the droplet, and other proteins were also observed (Fig. 1b). We analyzed the resulting peptide mixtures using LC–MS/MS and totally identified 4436 proteins with 99% certainty within three biological replicates (Supplementary Data 1). Of these proteins, 4204 and 4258 were identified with MS2 spectrum evidence in Mix and FT, respectively (Fig. 1c and Supplementary Fig. 1b). In the droplet, there were 1686 proteins and most of them overlapped with those in mixture and FT (Fig. 1c). As expected, mCherry-p62 and ubiquitin were strongly enriched as the top hits in the droplet fraction (Fig. 1b, d). The Pearson correlation coefficients (R) of protein abundance between replicates were higher than 0.9, and the mixture and FT were highly correlated but less with the droplet (Supplementary Fig. 1c). This indicates that the protein composition and abundance in mixture were highly similar to FT but significantly different to that of droplet. We then investigated the global diversity of proteomics samples by unsupervised hierarchical clustering of all quantified proteomes. The hierarchical matrix was clearly divided into two main groups: one cluster comprising samples from Mix and FT; and another cluster belonging to droplet samples (Fig. 1e). Analysis of variance (ANOVA) test filtered out 590 proteins as the enriched ones in the droplet with high confidence (Supplementary Data 2). Gene ontology (GO) enrichment analysis of the droplet enriched proteins showed significant enriched terms of multiple vesicle-related processes such as "membrane trafficking", "vesicle organization" and "autophagy" (Fig. 1f and Supplementary Data 3). In addition, gene set enrichment analysis (GSEA) of normalized counts of all proteins in droplet compared with those in mixture using the MSigDB database further showed that multiple membrane-associated process or component gene sets were significantly enriched in droplet rather than Mix (nominal p value < 0.01) (Supplementary Fig. 1d–g). These gene sets contain a large protein family to mediate the fusion of vesicles with the target membrane, including exocytosis, or membrane-bound compartments (such as phagocytic vesicle). The top two enriched terms "Regulation of exocytosis" and "SNARE complex" were highlighted (Supplementary Fig. 1e, g).

We further characterized the p62 adjacent proteins in living normal rat kidney (NRK) cells by an engineered ascorbate peroxidase (APEX)-based proximity labeling coupled with mass spectrometry[40] ("Methods"; Supplementary Fig. 2a–d). A total of 241 proteins were enriched in the APEX labeling group as p62 adjacent proteins in vivo (Supplementary Fig. 2e and Supplementary Data 4), and a majority of them (158 out of 241) were also present in the droplet (Supplementary Fig. 2f), supporting a high agreement between the two strategies for p62-associated mass spectrometry. Consistently, Metascape enrichment network analysis for the overlapped genes between the droplet and APEX group showed significant enrichment of membrane-associated processes, such as "vesicle-mediated transport", "membrane trafficking" and "selective autophagy" (Supplementary Fig. 2g).

As a further validation, we employed immunofluorescence assay to examine the colocalization status of p62 body with multiple membrane/vesicle structures or protein hits retrieved from proteomics profiling. Significant colocalization was observed between p62 bodies and multiple vesicle-related RAB proteins, SNARE components, and various ATG proteins (Supplementary Fig. 3a–c). Taken together, these

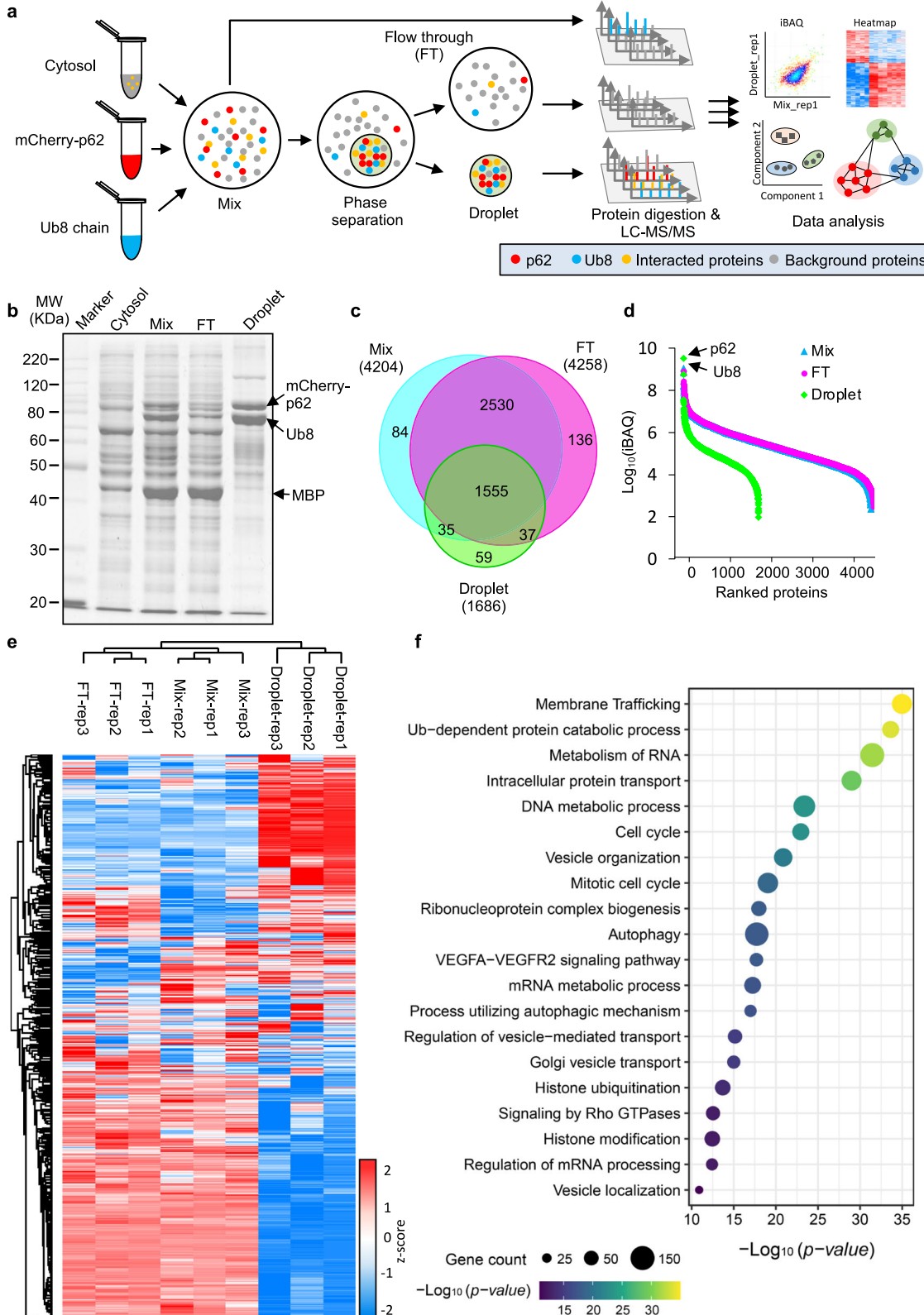

**Fig. 1 | Proteomics analysis showing that the p62-condensated body is highly associated with membrane proteins. a** Schematic workflow for the proteomics profiling of p62 body in vitro. The MS raw files were analyzed by MaxQuant and Perseus. **b** SDS-PAGE analysis of the cytosol, mixture (Mix), flow through (FT) and droplet. The black arrows indicate the mCherry-p62, Ub8 chain and the MBP tag derived from MBP-mCherry-p62 in the Mix, FT and droplet fractions. Three biological replicates of Mix, FT and droplet were performed and the similar trend was observed. **c** Venn diagram showing the overlap of proteins identified in the Mix, FT and droplet. **d** Dynamic range of the proteome abundance in the Mix, FT and droplet. **e** Unsupervised hierarchical clustering of all quantified proteomes in Mix, FT and droplet quantified proteomes. Three biological replicates were performed. **f** Metascape enrichment of proteins enriched in the droplet ($n = 590$). The top 20 enriched main cluster terms of biological pathways were selected and presented.

efforts systematically reveal the protein compositions of p62 bodies and suggest that p62 body tend to recruit membrane-associated proteins and possibly exert functions involving membrane-related mechanisms.

## p62 body nucleates core autophagy machinery and gathers small vesicles as autophagosome membrane sources

To further decipher the detailed mechanisms by which p62 bodies act during autophagosome formation, we examined the subcellular localization of p62 and core autophagy machineries. In starved normal rat kidney (NRK) epithelial cells, large p62 bodies, but not small p62 bodies, were significantly colocalized with FIP200, ATG16L1 (both total and phosphorylated form), ULK1, WIPI2 and DFCP1 (a marker for ER derived protrusion called omegasome) (Fig. 2a, b and Supplementary Fig. 4a, b), consistent with our previous report that only large p62 bodies are colocalized with LC3 and then subjected to autophagic degradation[38]. While less or none colocalization was observed with ATG9A, ATG14 (Supplementary Fig. 5a, b). These data indicate that p62 bodies may contain or recruit additional core autophagy machineries or complexes to promote autophagosome formation. Consistently, we indeed observed these proteins such as ATG2A/B, ATG3, ATG13, ATG101 and ATG16L1 within p62 droplet in our mass spectrometry and immunofluorescence data (Supplementary Data 1 and Supplementary Fig. 3b).

Considering the proteomics results and broad colocalization of p62 body with core components of autophagosome complexes and PI3P-bound membrane effector WIPI2, we posited that membrane structure and membrane-associated components in p62 body may play a role in autophagosome formation. To test this, we generated a *Atg2a/2b* double knockout (*Atg2ab DKO*) NRK cell line (Supplementary Fig. 5c), in which phospholipid supply and membrane fusion are blocked and autophagosome is arrested (Supplementary Fig. 5d), and then observed the localization of p62 bodies and core autophagy proteins in these cells. Interestingly, we found a significant difference between wild type (WT) cells and *Atg2ab DKO* cells. In WT cells, core autophagy proteins such as LC3, FIP200, WIPI2 and ATG14 were all interspersed inside p62 bodies; however, they were excluded from the core of p62 droplets and concentrated at the periphery of p62 bodies in cells. Other autophagic proteins such as LC3 and ATG14 rather exhibited a more flattened colocalization pattern with p62 puncta (Fig. 2c, d and Supplementary Fig. 5e–h). Such unexpected composition and architecture of p62 bodies implies a disordered autophagosome complex assembly and aberrant autophagosome biogenesis. Indeed, in *Atg2ab DKO* cells, a large number of membrane vesicles accumulated around the p62 body as shown by trans electron microscopy (TEM) (Supplementary Fig. 5d). Notably, these vesicles were positive with ATG9A or LC3 in *Atg2ab DKO* cells as evidenced by either confocal microscope or APEX-TEM (Fig. 2e, f). These small vesicular structures represent primary membrane sources for phagophore assembling and the failure of their fusion then led to arrested autophagosome formation in the absence of Atg2a/2b. It indicates that membrane structure and/or membrane-associated components gathered by p62 body are prerequisite for appropriate autophagosome biogenesis.

## p62 bodies are required for correct positioning of ATG9 and ATG16L1 vesicles

ATG9A- and ATG16L1-positive vesicles trafficking intracellularly around *trans*-Golgi network, endosome or plasma membrane were suggested to contribute membrane sources to early and/or growing phagophore[8]. Although p62 bodies did not significantly colocalized with ATG9A or partially colocalized with p-ATG16L1 vesicles in WT cells, they became more overlapped in *Atg2ab DKO* cells (Fig. 3a, b), suggesting that these Atg9 vesicles can transiently traffic around

p62 bodies to provide initial membrane precursors of phagophore. Without ATG2-mediated membrane tethering and lipid transfer, these small ATG9 vesicles failed to coalesce into large isolation membrane and got stuck around p62 bodies. To further explore the link between p62 bodies and membrane sources during autophagosome formation, we generated *p62* knockout NRK cells (Supplementary Fig. 6a) and examined the effects of p62 deficiency on the distribution and recruitment of membrane sources. The formation of autophagosome upon starvation was not drastically impaired in *p62* KO cells as shown by TEM and confocal microscopy (Supplementary Fig. 6b–d). This is likely due to the non-selective macroautophagy induced by starvation being more dependent on calcium transients as reported very recently[41], while p62 as a cargo receptor especially for ubiquitinated substrates is more essential during selective autophagy[32,42,43]. Indeed, we found that ubiquitination-modified proteins were more strongly accumulated in *p62* KO cells than WT cells upon starvation (Supplementary Fig. 6e), suggesting a compromised autophagic degradation of selective ubiquitinated cargos without p62. In *p62* KO cells, an aberrant accumulation of ATG9A-positive tubulovesicular structure around perinuclear region and increased puncta of p-ATG16L1 in the cytoplasm were observed even under nutrient-replete condition (Fig. 3c, d), indicating that p62 is required for proper positioning of ATG9A- and ATG16L1-positive vesicles. Further investigation showed that the re-located ATG9A significantly co-stained with Golgi marker but not with ER and less with ER-Golgi intermediate compartment (ERGIC) marker (Supplementary Fig. 7a). Other ATG proteins such as FIP200, DFCP-1, ATG16, ATG14, WIPI-2 and LC3 did not co-stain with ATG9A in p62 KO cells (Supplementary Fig. 7b). These data suggest that Golgi might be a primary source or relay station of ATG9A vesicles and without p62 body guidance these vesicles tend to be stuck in such position and fail to incorporate into growing autophagosome. Such aberrant structure and dislocation of ATG9A and ATG16L1 vesicles were different from those in *Atg2ab DKO* cells as in latter case ATG9A and ATG16L1 vesicles were accumulated in the cytosol as individual puncta (Fig. 3a, c). Moreover, the expression levels of ATG9A were significantly reduced in *p62* KO cells under both normal and starved conditions (Fig. 3e, f). These data indicate that p62 bodies can affect the positioning and abundance of ATG9A or ATG16L1 vesicles. This observation was further confirmed by APEX-TEM. Using APEX labeling, we observed a large number of ATG9A-labeled vesicles localized in the perinuclear region of *p62* KO cells (Fig. 3g).

To further determine the causal role of p62 bodies in recruiting membrane vesicles, we expressed either WT or p62 M404V mutant that fails to be phase separated in *Atg2ab* DKO cells[28] and found that such mutant did not recruit ATG9A and ATG16L1 vesicles (Supplementary Fig. 8a). Similar findings were also shown in p62 KO cells (Supplementary Fig. 8b, c). These data indicate that it is the p62 body but not diffused form of p62 that can recruit membrane vesicles. As p62 protein can directly interact with LC3 to assemble LC3-decorated vesicles, we next examined whether p62 bodies can still recruit membrane vesicles without direct LC3 contribution. Using a LIR mutant p62 (W338A, L341A) that does not bind to LC3[32], we observed that this mutant could still form droplet and was also capable to recruit ATG9A and ATG16L1 vesicles as well as core autophagosome machineries as evidenced by immunofluorescence and immunoprecipitation analysis (Supplementary Fig. 9a–c). These results further corroborate the importance of phase separated p62 bodies in recruiting membrane vesicles during autophagosome formation. Moreover, we used 1,6-hexanediol (known as a chemical to disrupt LLPS assemblies) to quickly dissolve p62 bodies and examined their re-formation after the chemical was washed off (Fig. 3h). We noticed that in *Atg2ab* DKO cells ATG9A localized to the perinuclear region accompanied with rapid dissolution of p62 bodies after treating with 1,6-hexanediol and the colocalization of ATG9A and p62 increased as 1,6-hexanediol was

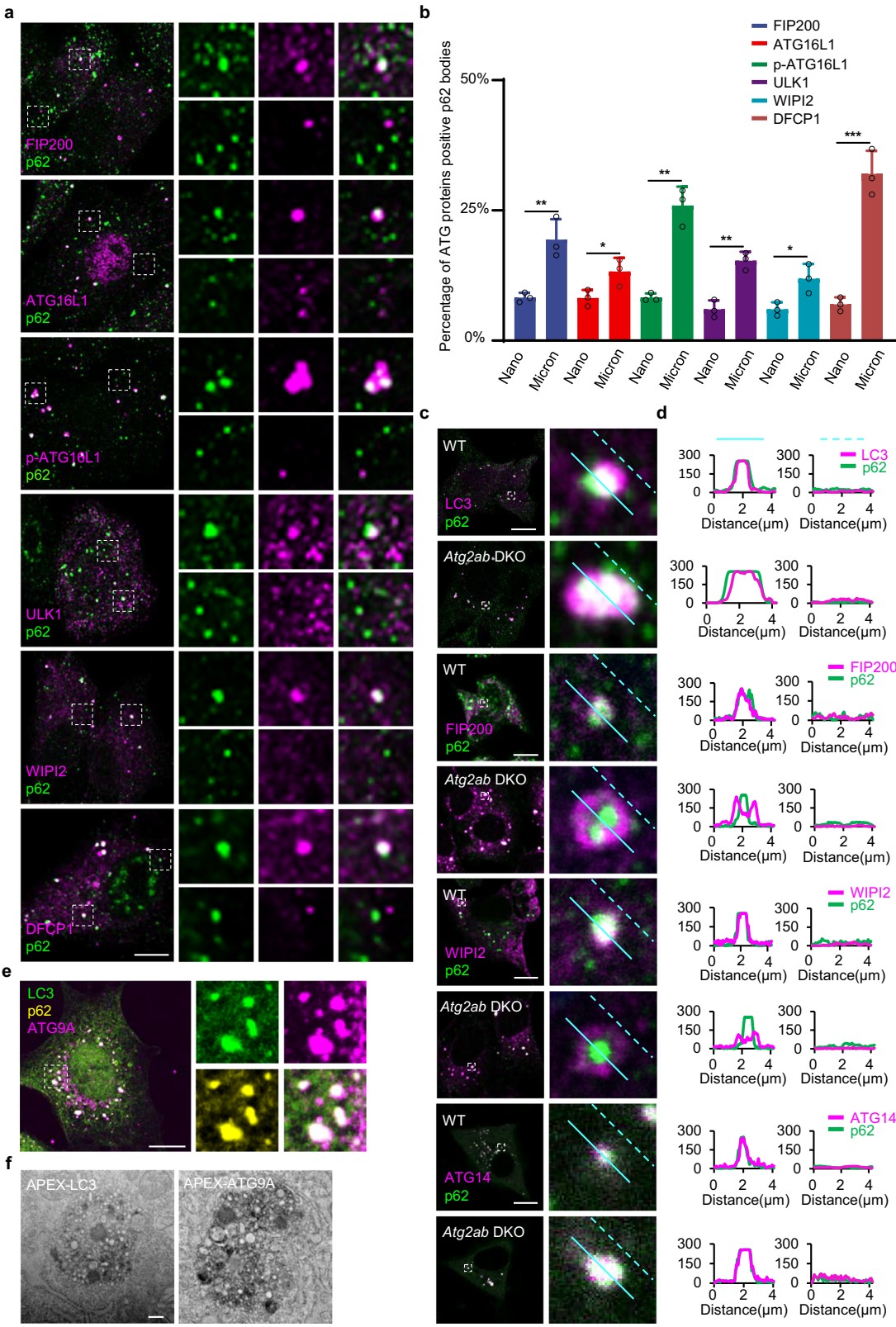

washed out (Fig. 3i and Supplementary Fig. 10a), indicating the direct positioning control of ATG9A vesicles by p62 bodies. Using similar assay, we also found that FIP200 condensate apparently emerged surrounding p62 bodies (Supplementary Fig. 10b), suggesting that p62 body serves as a nucleation center to establish FIP200 puncta spatially.

## p62 bodies recruit lipid membrane in vitro and in vivo

To further determine the relationship between p62 bodies and membrane structures, we reconstituted p62 bodies in vitro using purified p62 protein and poly-ubiquitin (Ub8) chain, and co-incubated them with extracted cytosol from *Atg2ab* DKO cells (Fig. 4a). Immunoblot analysis of either soluble fraction (S) or precipitate fraction (P) showed

**Fig. 2 | p62 body nucleates core autophagy machinery and recruits small vesicles as membrane sources for autophagosome generation. a** EGFP-tagged ULK1 or DFCP1 was transiently expressed in NRK cells. The cells were then starved for 4h and stained with antibodies against GFP and p62. NRK cells were starved for 4 h, and then stained with antibodies against FIP200, ATG16L1, p-ATG16L1, WIPI2 and p62. The left panel shows the co-localization of FIP200, ATG16L1, p-ATG16L1, ULK1, WIPI2, DFCP1 with p62. Right panels show enlarged p62 structures that were either FIP200, ATG16L1, p-ATG16L1, ULK1, WIPI2 and DFCP1 positive (upper panels) or FIP200, ATG16L1, p-ATG16L1, ULK1, WIPI2 and DFCP1 negative (upper or lower panels). Scale bar, 10 μm. **b** The percentage of FIP200, ATG16L1, p-ATG16L1, ULK1, WIPI2 and DFCP1 positive puncta in micron-scale (Micron) or nanoscale (Nano) p62 bodies quantified in cells from (**a**). Data are presented as mean ± SD, n = 3 independent experiments; 100 puncta were assessed per independent experiment. *p* values were calculated using the two-tailed, unpaired *t*-test. *$p < 0.05$, **$p < 0.01$, ***$p < 0.001$. **c** *Atg2ab* DKO and NRK cells were starved for 4 h, and then stained with antibodies against FIP200, WIPI2, LC3, ATG14 and p62. Scale bars, 10 μm. **d** Line profiling of a representative section of the cell, indicated by the blue lines in (**c**). **e** *Atg2ab* DKO cells were stained with antibodies against p62, ATG9A and LC3. Scale bars, 5 μm. **f** TEM image showing the DAB staining pattern in *Atg2ab* DKO cells transiently transfected with APEX2-LC3 or APEX2-ATG9A. Scale bar, 1 μm.

that FIP200, ATG16L1, ATG9A and LC3 could co-precipitate with p62 bodies, but not with p62-ΔPB1 which lacks self-oligomerization domain and cannot undergo phase separation (Fig. 4b). Similar results were also obtained by immunofluorescence analysis (Fig. 4c and Supplementary Fig. 11a–d). To exclude the effect of endogenous p62, we also performed sedimentation assay using *p62* knockout cell cytosol and similar conclusion was drawn (Supplementary Fig. 11e, f). To test whether p62 bodies can directly interact with lipid membrane, we employed a fluorophore-PE labeled liposome and co-incubated it with p62 bodies (Fig. 4d). Indeed, p62 bodies can significantly recruit liposome around the outer rim of droplet sphere (Fig. 4e), whereas other phase separated droplets such as SUMO, FUS, DDX4 and HURNPAB proteins had dramatically lower capability to attract this liposome (Supplementary Fig. 12a, b). These data suggest that the recruitment of liposome is specific to p62 bodies rather than general phase separated systems. This recruitment was further corroborated by TEM (Fig. 4f). To further determine which lipid contents are recruited by p62 bodies in vivo, we performed a lipidomic profiling of p62 droplet by mass spectrometry in both WT and *Atg2ab* DKO cells (Supplementary Data 5). The predominate lipids bound with p62 droplets were phosphatidylcholine (PC) and phosphatidylethanolamine (PE) (Supplementary Fig. 13a, b), which are known to be major membrane components of autophagosome[8,44]. Interestingly, the level of sphingomyelin (SM) of p62 body was dramatically increased in *Atg2ab* DKO cells (Supplementary Fig. 13a, b), consistent with the reports that sphingomyelin overload correlates with dysfunctional autophagosome formation[45,46]. These data collectively suggest that the formation of p62 droplets helps to direct the positioning and shaping of membrane structures. Such effect may be important for p62 bodies as the nucleation site for autophagosome formation.

### p62 bodies spatially regulate kinase reaction for autophagosome formation

Next, we purified ULK1 and PI3KC3-C1 complex and employed similar assays as above to determine their partition into the reconstituted p62 bodies (Supplementary Fig. 14a, b). As expected, ULK1 could evenly incorporated into p62 bodies (Fig. 5a, b). Interestingly, although PI3KC3-C1 was also co-sedimented with p62 bodies, this complex assembled at the outer rim of p62 droplet. Given that PI3KC3-C1 is primarily localized on the lipid membrane to catalyze the reaction for producing critical signal messenger PI3P, we surmised that the interaction between p62 body and PI3KC3-C1 might be vital for lipid phosphorylation. To determine the functional consequences of p62 body-mediated recruitment of PI3KC3-C1, we did the in vitro kinase assay by co-incubating in vitro reconstituted p62 bodies with purified PI3KC3-C1 complex and ULK1 in the presence or absence of ATP and analyzed the condensate or supernatant fraction by immunoblot assay (Fig. 5c, d). As core components of PI3KC3-C1, BECN1 and ATG14 can be phosphorylated by ULK1 in the presence of ATP, and such phosphorylation promotes PI3KC3-C1 activity and subsequent autophagosome formation[47–50]. In test tubes with purified proteins, we found that the co-sedimentation of ULK1, Beclin1 and ATG14 with p62 bodies was not significantly affected by the presence or absence of ATP (Fig. 5d, e).

In contrast, the phosphorylation level of Beclin1 (p-BECN1) was dramatically increased while to a lesser extent for p-ATG14 (Fig. 5d, e), suggesting that p62 body can support phosphorylation of PI3KC3-C1 by ULK1 at p62 body. To determine whether these recruited PI3KC3-C1 is active in vitro and contributes to PI3P generation on the membrane, we added liposome and PI3P-binding probe mCherry-FYVE into the reconstituted p62 bodies and PI3KC3-C1 in the presence or absence of ATP (Fig. 5f). Indeed, the addition of ATP promoted the puncta formation of mCherry-FYVE around the outer rim of p62 droplet whereas no such puncta were generated without PI3KC3-C1 (Fig. 5g, h). These in vitro assays clearly showed that the assembly of core autophagosome machineries around p62 body can spatially regulate kinase activities, leading to local PI3P generation and membrane tethering for accelerated autophagosome formation.

## Discussion

How membrane sources are organized together to initiate autophagosome formation has been a challenging question in this field for a long time. In this study, we systematically profiled the protein composition of p62 bodies with mass spectrometry-based proteomics. It gave us large-scaled information about the potential substrates undergoing p62 body-mediated autophagy. Moreover, we found a significant enrichment of autophagy machinery proteins, especially the membrane-associated proteins with proteomics, which indicates p62 body as a platform for autophagosome machinery organization and membrane source gathering. We further confirmed that, in addition to directly interact with FIP200 and LC3, the phase separated p62 bodies can act as a nucleation site to assemble multiple autophagosome complexes such as ULK1 complex and PI3KC3-C1 as well as multiple membrane sources including ATG9A- and ATG16L1-positive vesicles, thereby providing an efficient catalytic platform to produce PI3P and membrane tethering. Interestingly, we also found that p62 bodies are important in proper distribution of ATG9A- and ATG16L1-positive vesicles, suggesting another mechanism on how membrane precursors are gathered to form phagophore. Given that actin-based cytoskeletal network is essential for p62 body assembly[38], it is likely that the interplay between cytoskeleton and p62 bodies underlies the positioning control of ATG9A- and ATG16L1-positive vesicles.

Autophagosome formation requires a series of intricately regulated processes including core scaffold complex assembly, membrane precursor recruitment and modification, membrane expansion and shaping, and pore closure[7,8]. During non-selective autophagy, calcium transients on the ER surface trigger liquid-liquid phase separation of FIP200 to specify autophagosome initiation sites[41]. In selective autophagy, autophagosome biogenesis is initiated by the formation of p62 bodies through phase separation with p62 and polyubiquitinated cargos, and followed by the formation of autophagosome around p62 body and later delivered to lysosome for degradation. Our findings nicely bridge the gap between membraneless condensates and formation of membrane structure. Firstly, p62 body enriches ubiquitinated cargos through phase separation. Then, membraneless p62 body locally recruits autophagosome machinery and gathers

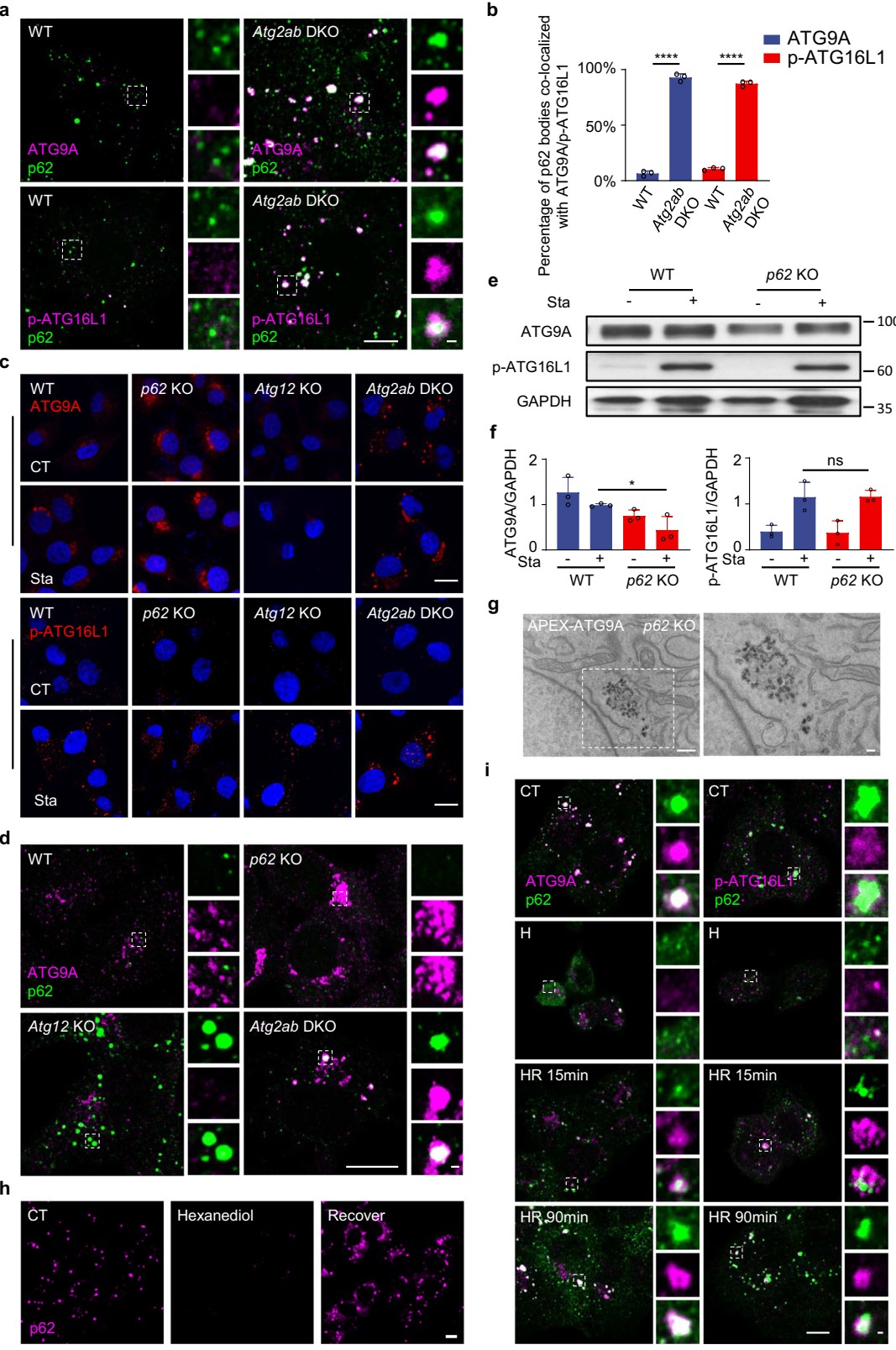

membrane sources to start the double membrane structure formation with ULK1 and invigorate PI3KC3-C1 kinase cascade for PI3P generation spatially around p62 body (Fig. 6). Such mechanism further expands the ways by which p62 participates in autophagosome biogenesis and deepens our understanding of the spatiotemporal control of membrane dynamics by biomolecular condensates in autophagy.

In summary, our study reveals a lipid membrane-centered mechanism by which p62 bodies exert functions in autophagosome formation. This action mode complements known mechanisms, demonstrating that p62 bodies affect multiple steps of autophagosome biogenesis from cargo segregation to double membrane structure formation. Further efforts are needed to unravel the details on the

**Fig. 3 | p62 body serve as a platform for the correct localization of ATG9 and ATG16L1 vesicles. a** NRK and *Atg2ab* DKO cells were fixed in paraformaldehyde and then stained with ATG9A, p-ATG16L1 and p62 antibodies. Co-localization of ATG9A, p-ATG16L1 and p62 was observed. Scale bar, 10 μm. **b** The percentage of ATG9A-, p-ATG16L1-positive p62 bodies was quantified in cells from (**a**). Data are presented as mean ± SD, *n* = 3 independent experiments; 100 puncta were assessed per independent experiment. *p* values were calculated using the two-tailed, unpaired *t*-test, ****$p$ < 0.0001. **c** NRK, *p62* KO, *Atg12* KO and *Atg2ab* DKO cells were treated without (CT) or with starvation (STA) using DMEM medium for 4 h, and then stained with antibodies against ATG9A and p-ATG16L1. Scale bars, 10 μm. **d** NRK, *p62* KO, *Atg12* KO and *Atg2ab* DKO cells were starved with DMEM medium for 4 h, and then stained with antibodies against ATG9A and p62. Scale bars, 10 μm. **e** NRK cells or *p62* KO cells were starved with DPBS (STA) for 4 h, and then the cell lysates were analyzed by immunoblotting with antibodies against ATG9A,

p-ATG16L1 and GAPDH. **f** The ratio of ATG9A or p-ATG16L1 to GAPDH were analyzed from (**e**). Data are presented as mean ± SD, *n* = 3 independent measurement of band intensity, *p* values were calculated using the two-tailed, unpaired *t*-test, *$p$ < 0.05. **g** TEM image showing the DAB staining in *p62* KO cells transiently transfected with APEX2-ATG9A. Scale bar, 1 μm. **h** Reversible effect of 1, 6-hexanediol on the formation of p62 bodies. *Atg2ab* DKO *cells* were starved for 2 h and cells were treated with 2% 1, 6-hexanediol for 20 min (H). Next the cells were recovered by washing with PBS and incubation with whole medium (HR) for 2 h. Then cells were stained with an antibody against p62. Scale bar, 10 μm. **i** *Atg2ab* DKO cells were starved for 2 h and cells were treated with 2% 1, 6-hexanediol for 20 min (H). Next the cells were recovered by washing with PBS and incubation with whole medium (HR) for 15 min or 90 min. Then cells were stained with an antibody against p62 and ATG9A or p-ATG16L1. Scale bar, 10 μm.

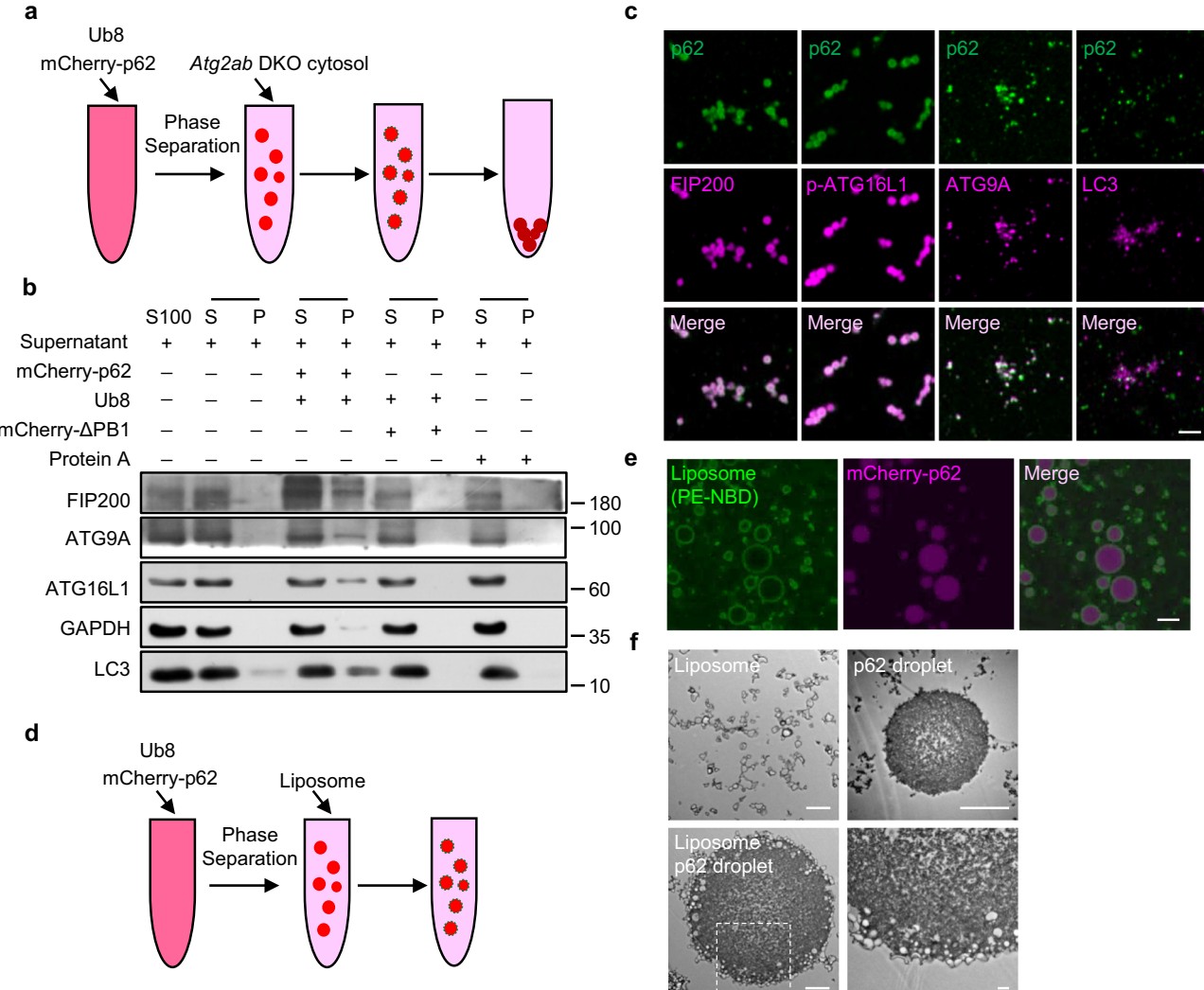

**Fig. 4 | In vitro recruitment of lipid membrane by p62 bodies. a** Schematic diagram of the sedimentation assay to separate the condensed liquid droplets and the supernatant. Phase separation assay of mCherry-p62 with linear Ub8 in vitro. *Atg2ab* DKO cytosol was added after the phase separation. mCherry-p62, 6 μM; His-Ub8: 2 μM; △PB1, 6 μM. **b** The sedimentation and supernatant from (**a**) were separated by centrifugation and analyzed by western blot using antibodies against FIP200, ATG9A, ATG16L1, LC3 and GAPDH. S: supernatant, P: pellet. **c** p62 body and different protein-interacting vesicles from (**a**) were co-stained with antibodies

against p62, FIP200, ATG9A, ATG16L1 and LC3. Scale bar, 2 μm. **d** Schematic diagram of the sedimentation assay to separate the condensed liquid droplets and the supernatant. Phase separation assay of mCherry-p62 with linear His-Ub8 in vitro. Liposome was added after the phase separation. mCherry-p62, 50 μM; His-Ub8: 15 μM; Liposome, 1 μM. **e** Liposomes (PE-NDB) around the p62 bodies were observed from the (**d**). Scale bar, 1 μm. **f** Liposomes (PE-NDB) around the p62 bodies were observed by TEM from the (**d**). Scale bar, 500 nm.

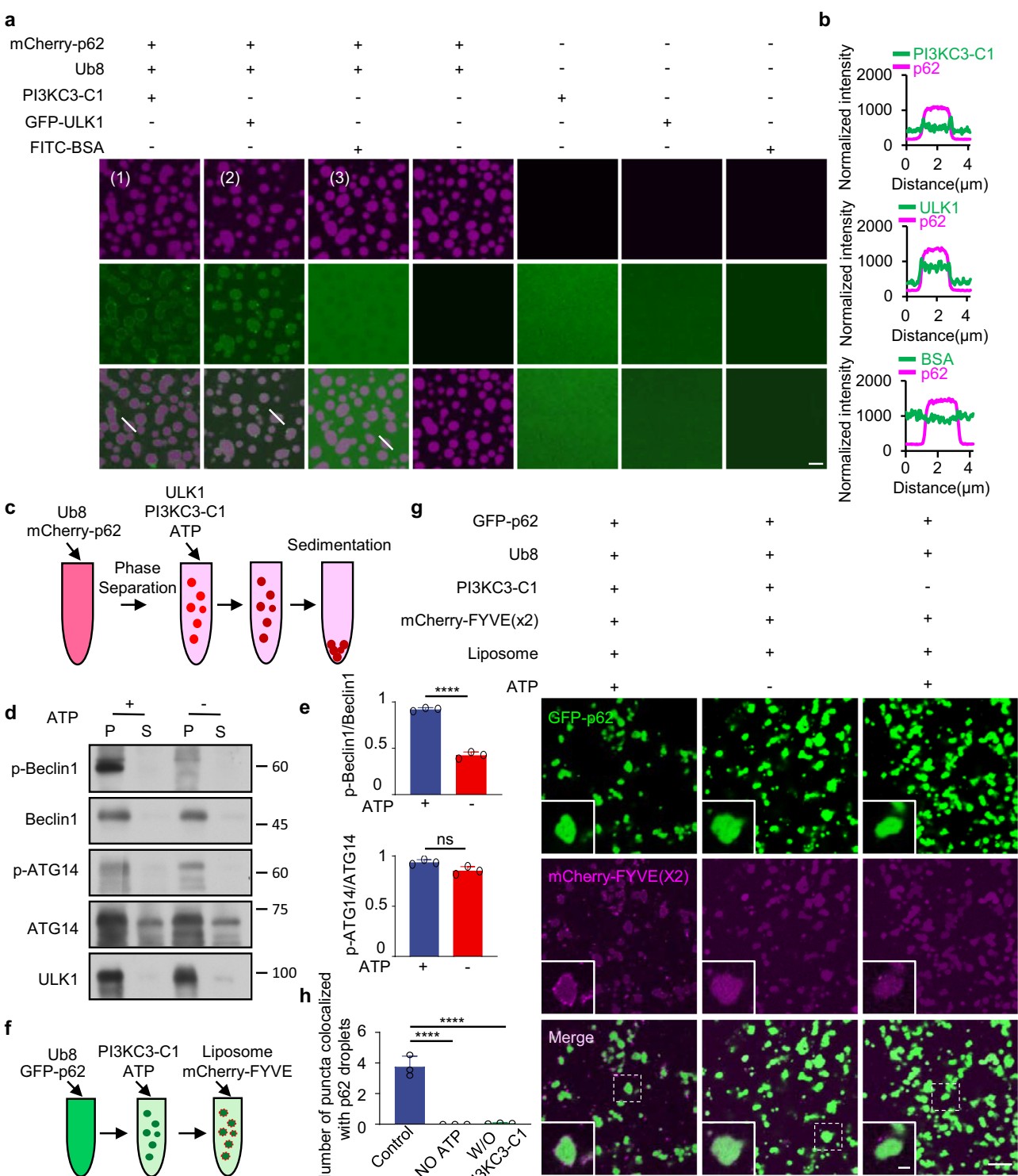

**Fig. 5 | p62 bodies regulate autophagosome formation by a spatially concentrated PI3K kinase subunit. a** TIRF images showing that droplets formed by LLPS of mCherry-p62 (50 μM), His-Ub8 (15 μM), PI3KC3-C1 (GFP-Beclin1) and GFP-ULK1 (1 μg/μl) fused upon contact in vitro. Scale bar, 5 μm. **b** Line profiling of a representative section of the p62 droplets, indicated by the white lines in panels (1)–(3) from (**a**). **c** Schematic diagram of the sedimentation assay to separate the condensed liquid droplets and the supernatant. In vitro phase separation assay of mCherry-p62 with linear His-Ub8. ULK1, PI3KC3-C1 and ATP were added after the phase separation. mCherry-p62, 50 μM; His-Ubx8: 15 μM; ULK1, (1 μg/μl); PI3KC3-C1, (1 μg/μl); ATP, 10 μM. **d** The sedimentation and supernatant were separated by centrifugation and analyzed by western blot using antibodies against p-Beclin1, Beclin1, p-ATG14, ATG14 and ULK1. S: supernatant, P: pellet. **e** The ratio of p-Beclin1

or p-ATG14 to total Beclin1 or ATG14 were analyzed from (**d**), Data are presented as mean ± SD, $n = 3$ times measurement of band intensity, $p$ values were calculated using the two-tailed, unpaired $t$-test, ****$p < 0.0001$. **f** In vitro phase separation assay of GFP-p62 with linear His-Ub8, and then PI3KC3-C1 and ATP were added up to the phase separation system. Liposome and mCherry-FYVE were finally added into the above phase separation system. GFP-p62, 50 μM; His-Ub8: 15 μM; PI3KC3-C1, (1 μg/μl); ATP, 10 μM; Liposome, 1 μM; mCherry-FYVE, 1 μM. **g** Liposomes (PE-NDB) and mCherry-FYVE(X2) around the phase separated p62 bodies were observed from (**c**). Scale bar, 1 μm. **h** The number of mCherry-FYVE puncta colocalized with p62 droplet was quantified. Data are presented as mean ± SD, $n = 3$ independent experiments. w/o means without. $p$ values were calculated using the two-tailed, unpaired $t$-test. ****$p < 0.0001$.

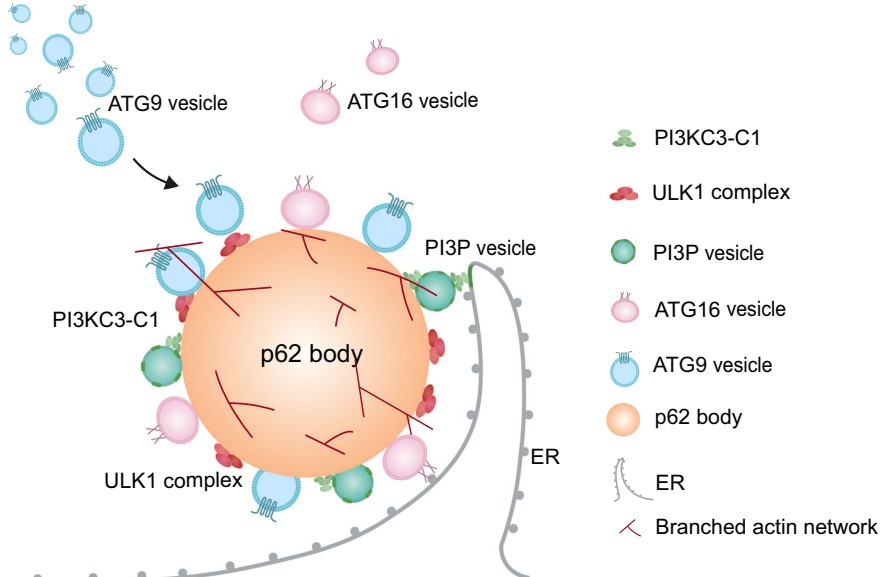

**Fig. 6 | Model for p62 bodies in setting the nucleation sites for autophagosome formation.** p62 body enriches ubiquitinated cargos through phase separation, then locally recruits autophagosome machinery such as ULK1 complex and gathers membrane sources including ATG9- and ATG16-positive vesicles to promote the double membrane structure formation. p62 body also invigorates PI3KC3-C1 kinase cascade for PI3P generation spatially around membrane-enriched p62 body to enhance downstream signaling cascade for autophagosome formation. This model reveals a lipid membrane-centered mechanism of p62 bodies in setting the nucleation sites for autophagosome formation.

biophysical interactions between p62 bodies and different membrane sources.

## Methods

### Cell culture, transfection and plasmids

The HEK239T and NRK cells were obtained from the American Type Culture Collection (ATCC). Cells were cultured in Dulbecco's Modified Eagle Medium (DMEM, Life Technologies; # 11960) with 10% fetal bovine serum (FBS, BI; # 04-002-1A-US), 1% glutamine (Glu, Gibco; #A2916801), 50 µg/ml penicillin/streptomycin and maintained at 37 °C under 5% $CO_2$. For treatment, cells were washed three times with phosphate-buffered saline (PBS), and incubated in starvation DMEM medium (Life Technologies; # 11960) or Dulbecco's Phosphate-Buffered Saline (DPBS, Gibco; # 14287) for the indicated time period. The Cells were transfected with appropriate dosage of DNA constructs via Amaxa nucleofection using solution T and program X-001. The p62 constructs were generated by cloning human p62 (NCBI Accession number: 8878) into pEFGP-C3 vector. The MBP-mCherry-p62 constructs were generated by cloning human p62 into His6-MBP-Tev-mCherry vector. The *p62* knockout NRK cell (*p62* KO), *Atg12* knockout NRK cell (*Atg12* KO), Atg2ab double knockout NRK cell (*Atg2ab* DKO) and plasmids (GFP-p62WT, M404V, LIR mut, GFP-DFCP1, GFP-ATG14) were gifts from Dr. Li Yu's Lab. EGFP-tagged RAB5B, RAB5C, RAB1A, RAB2A, STX17, STX6, STX12 and STX3 plasmids were gifts from Dr. Yueguang Rong. EGFP-SEC61β and EGFP-ERGIC53 plasmids were gifts from Dr. Liang Ge. SUMO, FUS, DDX4 and HNRNPAB proteins were kindly provided by Dr. Pilong Li.

### p62 droplet preparation

All proteins were expressed in *Escherichia coli* BL21 Rosetta (DE3) cells. For maltose binding protein (MBP)-mCherry-p62 and His6-Ub8, cells were re-suspended in lysis buffer (150 mM NaCl, 40 mM Tris-HCl pH 7.4, 10% glycerol, 4 mM Dithiothreitol (DTT), 1 mM EDTA, protease inhibitor cocktail). MBP-tagged mCherry-p62 was purified through a MBPTrap HP column. His-tagged Ub8 was isolated through a Ni-NTA column. We isolated S100 fraction of cytosol from HEK239T suspension cells (50 µg/µl) and mixed with mCherry-p62 protein (100 µM) as a

ratio of 1:1, followed by adding the same volume of His-Ub8 (30 µM) to form p62 droplets at room temperature for 3–5 min. An aliquot of this cytosol, p62 and His-Ub8 mixture was taken out as a mixture (Mix) sample, and the left was centrifuged at $12,000 \times g$ for 5 min. The supernatant was taken out as a flow through (FT) sample. The pellet was washed gently three times with phase separation buffer (150 mM NaCl, 40 mM Tris-HCl pH 7.4, and 1 mM DTT) to reduce the contaminant proteins. The pellet was used as the droplet sample for identifying the client protein partitioned into p62 body. All the samples were boiled at 80 °C for 5 min with 1x loading buffer and used for next step MS detection.

### APEX2-based proximity labeling

For APEX2-GFP-p62 labeling, the plasmid was transfected by using Vigofect Reagent (Vigorous Biotechnology, T001) to NRK cells with a confluency up to ~ 60–70%. After 24 h, change the fresh DMEM medium with Biotin-phenol (BP, 500 µM, Aladdin, B288371) and culture for 60 min. Three replicates were independently performed for control and APEX labeling groups. The label groups were added with 1 mM $H_2O_2$ for 1 min and control groups were added with equal volume of PBS. APEX activity was extinguished with freshly quenching solution (QS: sodium azide (10 mM, Amresco, 0639), sodium ascorbate (10 mM, Sigma-Aldrich, A7631) and Trolox (5 mM, Aladdin, T488850) in PBS) for two times and then twice with cold PBS. Cells were then scraped in PBS, centrifuged at $800 \times g$ for 3 min at 4 °C. The cell pellet was lysed in ice-cold RIPA lysis buffer (50 mM Tris, 150 mM NaCl, 0.1% SDS, 0.5% sodium deoxycholate and 1% Triton X-100 in Millipore water) supplemented with N-Ethylmaleimide (NEM, 1 mM, Sigma-Aldrich, E3876) and phenylmethylsulfonyl fluoride (PMSF, 1 mM, Amresco, 0754). Lysates were centrifuged at $15,000 \times g$ for 5 min at 4 °C. Protein concentration was quantified with Pierce BCA Protein Assay Kit (Thermo Fisher, 23227). About 1 mg protein was used for enrichment analysis. Streptavidin-coated beads (Pierce Streptavidin Agarose Beads, Thermo Fisher, 20353) were incubated for pulldown experiments with 500 µl of the extract at a ratio of 30 µl beads per 1 mg of sample with rotation overnight at 4 °C. The beads were washed with 1 ml RIPA lysis buffer for 4 times,

urea wash buffer (Urea, 2 M in 10 mM Tris-HCl, pH 8.0) for 4 times and ABC wash buffer (50 mM ABC in 10 mM Tris-HCl, pH 8.0) for 4 times. Then the biotinylated proteins from the beads were eluted by boiling each sample in 30 μl of 1x protein loading buffer supplemented with 2 mM biotin and 20 mM DTT for 10 min.

### Sample preparation and LC-MS/MS analysis

The protein samples of mixture, flow through and p62 bodies or biotinylated samples were reduced with 5 mM dithiothreitol (DTT) at 60 °C for 15 min and alkylated with 15 mM iodoacetamide (IAM) at room temperature in darkness for 30 min. Protein lysates were separated by 10% SDS-PAGE and sliced into fractions based on sample complexity and molecular weight markers and digested in gel with 10 ng/μl trypsin (Enzyme & Spectrum, Beijing, China) dissolved in digested buffer (50 mM $NH_4HCO_3$ + 5% ACN) overnight at 37 °C. The tryptic peptides were extracted with extraction buffer (5% formic acid and 45% acetonitrile) and finally dried using a vacuum dryer (LABCONCO CentriVap, Kansas City, USA). The resulting peptides were analyzed on the LC-MS/MS platform of hybrid LTQ-Orbitrap Velos mass spectrometer (Thermo Fisher Scientific, San Jose, CA, USA) equipped with the ultra-performance liquid chromatography (UPLC) system (Thermo Fisher Scientific, San Jose, CA, USA). Peptides were separated on a 75 μm I.D. × 15 cm capillary column (Enzyme & Spectrum, Beijing, China) packed with 1.9 μm C18 reverse-phase fused-silica (Michrom Bioresources, Inc., Auburn, CA, USA). The LC nonlinear gradient with 100 min ramped from 8% to 40% of mobile phase B (phase B: 0.1% FA in ACN, phase A: 0.1% FA + 2% ACN in water) at a nano flow rate of 300 nl/min. The MS1 was detected with a mass range of 300–1600 at a resolution of 30,000 at $m/z$ 400. The automatic gain control (AGC) was set as $1 × 10^6$ and the maximum injection time (MIT) was 150 ms. The MS2 was detected in data-dependent mode for the 20 most intense ions subjected to fragmentation in the linear ion trap (LTQ). For the first replicate, the AGC was set at $1.5 × 10^4$ and the MIT was set at 50 ms. The dynamic range was set at 35 s to suppress repeated detection of the same ion peaks. For the other two replicates, the AGC was set at $1.0 × 10^4$, the MIT was set at 30 ms and the dynamic range was set at 50 s. The whole experiments were repeated for three times.

### MS data analysis for protein identification and quantification

All the raw files were searched by MaxQuant (version 1.5.38)[51] against the Swiss-Prot reviewed human database (version 2020.08, containing 20375 entry proteins). For proteome identification, searching parameters consisted of full tryptic restriction and peptides were allowed up to two miss cleavages. Precursor mass tolerance was set at 20 ppm for the first search and 6 ppm for the main search. The tolerance of MS2 fragments was set at 0.5 Da. Carbamidomethylation of cysteine was specified as a fixed modification, and oxidation of methionine and protein N-terminal acetylation were assigned as variable modifications. The spectrum, peptides and proteins were filtered with false discovery rate (FDR) lower than 1% using a target-decoy search strategy.

For quantification, at least two unique or razor peptides were required for protein quantification. The "match between runs" option was selected to transfer identification between separate LC-MS/MS runs based on their accurate mass and retention time after retention time alignment. The label-free quantification (LFQ) algorithm in MaxQuant was used to compare between Mix, FT and droplet samples. ComBat was used to adjust for batch effects between replicate one and the other two datasets as described[52]. Using the three biological replicate measurements, differentially (droplet enriched and excluded) included proteins were identified using Perseus (version 2.0.7)[53]. The LFQ values were transformed by log2, and protein groups were filtered to retain only proteins with at least 3 valid values. Missing data were imputed by creating an artificial normal distribution with a downshift of 1.8 standard deviations and a width of 0.3 of the original ratio distribution. $Z$-score was used to normalize the quantification value between Mix, FT and droplet. Then we used ANOVA test with Benjamini-Hochberg FDR ≤ 0.01, and clustered by Euclidean coefficients as the default parameters.

### Chemicals and reagents

Cells were incubated with fresh DMEM medium supplemented with 10% FBS or DMEM medium for 12–16 h. Formation of p62 bodies was induced by DMEM medium or DPBS. Cells were treated with 1, 6-Hexanediol (2%, Sigma-Aldrich, # 629-11-8) was added to *Atg2ab DKO* cells for 20 min.

### Immunoprecipitation and immunoblot analysis

EGFP-tagged p62 or EGFP-tagged p62 LIR Mut was transiently expressed in *p62* KO cells. Immunoprecipitation analysis was performed with IgG or GFP antibodies and protein A/G Magnet Beads (Selleck, #B23202). The interacting proteins were analyzed by immunoblotting with FIP200, ATG9A, GFP, ATG16L1, LC3 and GAPDH antibodies. For immunoblot analysis, cells were washed twice with ice-cold PBS, lysed in dishes with lysis buffer (50 mM Tris-HCl, pH 7.4, 2% SDS and 2 mM EDTA pH 8.0), and then mixed with 5 × SDS-PAGE loading buffer to final 1 × SDS-PAGE concentration (0.6 M Tris-HCl pH 6.8, 2% SDS, 25% glycerol, 14.4 mM DTT and 0.1% bromophenol blue) before boiling at 100 °C for 15 min. Lysates were separated on 10% SDS-PAGE gels. Proteins were transferred to nitrocellulose membrane (PALL NT nitrocellulose, # 66485) and blocked with 5% skim milk (BD, # 232100) in PBS for 2 h at room temperature, followed by antibody incubation. Rabbit anti-actin (Sigma-Aldrich, # A2066, 1:50,000), rabbit anti-p62 (MBL, # PM045, 1:1000), mouse anti-p62 (MBL, # PM162-3, 1:1000), rabbit anti-WIPI2 (abcam, # ab229225, 1:1000), mouse anti-ATG9A (proteintech, # 67096-1-Ig, 1:500), rabbit anti-ATG9A (abcam, # ab108338, 1:1000), rabbit anti-ATG16L1 (abcam, # ab188642, 1:1000), rabbit anti-p-ATG16L1 (abcam, # ab195242, 1:500), rabbit anti-p-ATG14 (CST, # 92340S, 1:1000), rabbit anti-ATG14 (MBL, # PD026MS, 1:1000), rabbit anti-p-Beclin1 (CST, # 84966S, 1:1000), rabbit anti-Beclin1 (abcam, # ab207612, 1:1000), anti-mCherry (abcam, # ab183628, 1:1000), rabbit anti-ULK1 (abcam, # ab240916, 1:1000) and mouse anti-ubiquitin (CST, # 3936S, 1:5000) were used as primary antibodies. Mouse IgG (beijing ding changsheng biotechnology company, #NIS-0041) was used as control antibody in Immunoprecipitation assay. Incubation was carried out for 1 h at room temperature, followed by four washes of 10 min each with PBST buffer. Then membranes were incubated with goat anti-rabbit or goat anti-mouse secondary antibody for 1 h at room temperature. After extensive washing for 40 min with PBST, membranes were finally exposed to the X-ray film.

### Immunostaining and confocal fluorescence microscopy

Cells were grown on coverslips. After treatment, cells were washed with ice-cold PBS, fixed in 4% paraformaldehyde for 10 min at room temperature, and then washed with PBS three times for 10 min each. Cells were perforated by incubating for 10 min in PBS containing 0.1% saponin. After blocking with goat serum in PBS for 30 min, cells were stained with antibody in blocking buffer for 1 h, and washed with PBS three times. The antibodies used were as follows: mouse anti-GFP (Roche, # 11814460001, 1:500), rabbit anti-p62 (MBL, # PM045, 1:1000), rabbit anti-LC3 (MBL, # PM036, # PM046, 1:500), mouse anti-p62 (MBL, # PM162-3, 1:500), rabbit anti-WIPI2 (abcam, # ab229225, 1:200), mouse anti-ATG9A (proteintech, # 67096-1-Ig, 1:100), rabbit anti-ATG9A (abcam, # ab108338, 1:200), rabbit anti-ATG16L1 (abcam, # ab188642, 1:200), rabbit anti-p-ATG16L1 (abcam, # ab195242, 1:50) and Mouse Anti-GM130 (BD,# 610822, 1:300). After cells were stained with conjugated secondary antibody in PBS for 1 h and washed with PBS three times at room temperature. Observe and capture the images with

a Nikon laser confocal microscope. Big p62 bodies were defined as those with a diameter larger than 300 nm while the diameter of small ones was under 300 nm. The quantification was performed by ImageJ.

### Electron microscopy

Cells cultured in 3.5 cm dishes were fixed by 1:1 (v/v) 2.5% glutaraldehyde and DMEM for 5 min at room temperature. The liquid was removed and the cells were incubated in 2.5% glutaraldehyde 45 min at 4 °C. The cells were then washed twice with 0.1 M sodium cacodylate buffer (pH 7.3) for 5 min at 4 °C, and treated with 20 mM glycine for 5 min at 4 °C. After three times washes for 5 min at 4 °C, the cells were incubated with 1 mg/ml DAB for 2 min at room temperature, then $H_2O_2$ was added to 0.03% (v/v) and the incubation was continued for 15 min. After three more washes, the cells were observed by electron microscopy (Hitachi 7650B HT7800 transmission electron microscope). All the solutions above were in 0.1 M sodium cacodylate buffer (pH 7.3).

### Cytosol preparation

*Atg2ab* DKO cells were digested with trypsin, centrifuged at $1670 \times g$ for 5 min, and the supernatant was discarded. The cells were washed once with an equal volume of cold PBS, and 3 times the volume of hypotonic buffer was added to the pellet to give a gentle suspension, and the cells were left on ice for 20 min. Next, the cell suspension was centrifuged at $1670 \times g$ for 5 min and the supernatant was discarded. Then, the isotonic lysis buffer was added to the cell precipitate and mixed well. The cell suspension was Dounce grinding 50 times and centrifuged at $3000 \times g$ for 10 min. The supernatant (cytoplasmic proteins) was transferred to a new cryotube, divided into fractions, frozen in liquid nitrogen and stored (long-term storage at −80 °C, short-term storage at −20 °C).

### In vitro protein and vesicles binding assay

Expression and purification of mCherry-p62 WT, ΔPB1 mCherry-p62, and Ub8 were described previously[28]. Pretreated mCherry-p62 WT + Ub8, ΔPB1 mCherry-p62 + Ub8, proteinA + Ub8 were added to a reaction system containing 90 μl of *Atg2ab DKO* cells cytosol and incubated for 20 min at room temperature. Next, the reaction system was centrifuged at $3000 \times g$ for 10 min. Finally, 10 μl of supernatant was taken for immunofluorescence and the remaining supernatant and pellet were immediately separated into separate tubes, each fraction being supplemented with precipitation buffer (150 mM NaCl, 40 mM Tris-HCl pH 7.4, 10% glycerol, 1 mM DTT) to give a final volume of 80 μl. For imaging, p62 droplets were co-stained with 1 μg/ml WGA-Alexa 488. Finally, the samples were analyzed by Western blot.

### Liposome preparation

Liposome was prepared as reported before. Briefly, lipid mixture with 55% DOPE, 35% POPC, 15% liver PI and 0.1% PE-NBD in chloroform was dried with nitrogen stream and further dried for 1.5 h at 37 °C. Phase separation buffer (150 mM NaCl, 40 mM Tris-HCl pH 7.4, and 1 mM DTT) was added to the lipid film and resuspended at 37 °C for 1 h on a shaker at the speed of $300 \times g$. Freeze and thaw was followed for 15 times until the lipid mixture became clear to form liposome. The liposome was further uniformed by extruding with a 1000 nm pore size polycarbonate film for 21 times.

### p62 droplet recruitment of liposome

One mM liposome was added to the p62 droplets formed with mCherry-p62 (50 μM) and His-Ub8 (15 μM) for 30 min. The recruitment of liposome to p62 droplets was imaged under confocal or electron microscope.

### Lipidomic profiling of p62 bodies

The recombinant mCherry-p62, Ub8 linear chain and cytosol from WT or *Atg2ab* DKO NRK cells were mixed to form p62 droplet, and the droplet was collected by centrifugation. Lipid analysis was conducted according to previous methods[54,55]. A Q Exactive Orbitrap mass spectrometer (Thermo, CA) was used for lipid analysis. Lipid separation was performed with an XSelect CSH C18 column at 45 °C. A binary solvent system consisting of mobile phase A (acetonitrile: $H_2O$, 60:40, 10 mM ammonium acetate) and mobile phase B (isopropanol: acetonitrile, 90:10) was utilized at a flow rate of 250 μl/min with the gradient scheme as follows: 0 min, 37% B; 1.5 min, 37% B; 4 min, 45% B; 5 min, 52% B; 8 min, 58% B; 11 min, 66% B; 14 min, 70% B; 18 min, 75% B; 20 min, 98% B. Data were acquired by LC-MS/MS analysis using Orbitrap. Mass resolution was set as 70,000 and 17,500 for MS and MS/MS acquisition. The top 10 of the most intense precursors were selected for fragmentation. Lipids were identified and quantified using LipidSearch 4.1.30 (Thermo, CA) and the chromatographic areas confirmed by Tracefinder 3.2 (Thermo, CA) were employed for relative quantitation.

### Partition assay

p62 droplet was formed with mCherry-p62 (50 μM) and His-Ub8 (15 μM). One μM client proteins including PI3KC3-CI (GFP-Beclin1), GFP-ULK1 and FITC-BSA (negative control) were then added to p62 droplets for 15 min. The partition of client proteins was imaged under confocal microscope.

### In vitro ULK1 kinase assay in p62 droplet

p62 droplet was formed with mCherry-p62 (50 μM) and His-Ub8 (15 μM) in phase separation buffer (150 mM NaCl, 40 mM Tris-HCl pH 7.4, and 1 mM DTT). One μg/μl purified PI3KC3-C1 and GFP-ULK1 was added to p62 droplets for 30 min with or without 10 μM ATP. The sample was centrifuged at $12,000 \times g$ for 5 min to separate the pellet and supernatant. Reaction was quenched by adding 1x SDS loading buffer to supernatant and pellet followed by boiling for 5 min. The samples were then separated by SDS-PAGE followed by western blot with phospho-Beclin1 and phospho-ATG14 antibody.

### In vitro VPS34 lipid kinase assay in p62 droplet

p62 droplet was formed with GFP-p62 (50 μM) and His-Ub8 (15 μM) in phase separation buffer (150 mM NaCl, 40 mM Tris-HCl pH 7.4, and 1 mM DTT). One μg/μl purified PI3KC3-C1 and 10 μM ATP was added to p62 droplet followed by adding 1 mM liposome (55% DOPE, 35% POPC, and 15% liver PI) and 1 μM mCherry-FYVE (x2). Sample was incubated at 37 °C for 30 min and imaged by confocal microscope.

### Statistics and reproducibility

Statistical analysis was performed using GraphPad Prism 8. Statistical analysis was carried out on the data from independent experiments. The error bars in the figures represent s.d. and the $n$ value is specified in the legends. Unless specified, each experiment was repeated independently for three times with similar results, and the representative results were shown.

### Reporting summary

Further information on research design is available in the Nature Portfolio Reporting Summary linked to this article.

## Data availability

The mass spectrometry data generated in this study have been deposited in iProX (http://www.iprox.org/) under accession code IPX0005450000 and also linked to the ProteomeXchange Consortium (http://proteomecentral.proteomexchange.org) with the dataset identifier PXD038262. Source data are provided with this paper.

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

## Acknowledgements

We are grateful to Dr. Li Yu and Dr. Pilong Li for critical reading and insightful suggestions about this work. We thank Drs. Xiaohui Liu and Lina Xu for technical support. We thank Metabolomics and Lipidomics Center of National Protein Science Facility, Tsinghua University for the technical support of lipidomic analysis. This work was supported by Xinjiang Uygur Autonomous Region Tianshan Talent Training Program (2022TSYCCX0030) to N.M.; the Ministry of Science and Technology of China (2020YFE0202200) and the National Natural Science Foundation of China (31771536 and 31860316) to N.M. and (32071431) to Y.L.; the CAMS Innovation Fund for Medical Sciences (2019-I2M-5-017) and Mass Spectrometry Platform Open Project of National Center for Protein Sciences Beijing (2021-NCPSB-001) to P.X.

## Author contributions

N.M. conceived the study; N.M. and D.S. wrote the manuscript with the input from other authors. N.M., D.S. and P.X. supervised the project; X.F., D.S., Y.L. and J.Z. performed most of the experiments. S.L., J.X.Z., Q.X., J.H., T.L., H.L., D.Z., M.X. and M.M. contributed to parts of the experiments. W.Z. and Y.L. helped with the electron microscopy. A.H. and M.M. provided insightful suggestions. All the authors discussed and commented on the manuscript.

## Competing interests

The authors declare no competing interests.
