## [Peer Review File · Nature Communications]

Local membrane source gathering by p62 body drives autophagosome formationReviewer #1 (Remarks to the Author):

Autophagy, a complex mechanism for the degradation of a variety of cellular constituents, relies on the spatio-temporal assembly of specific components forming double-membraned structures called autophagosomes. During biogenesis and membrane elongation, the forming autophagosome relies on constant membrane supply. Detailed mechanisms behind this process, especially with regard to p62, a ubiquitous receptor for autophagic cargo, remain elusive. In the present manuscript Feng et al. describe membrane-recruitment, and potentially underlying mechanisms, to phase-separated p62 bodies. Upon profiling of p62 body composition by LC-MS, the authors describe the presence of autophagic components and membrane sources (e.g. ATG9 positive vesicles) in p62 bodies. This was enhanced in ATG2 deficient cells. Secondly, the authors propose that p62 bodies constitute a platform for the assembly of ULK1 and PI3KC3-C1 complexes, thus initiating autophagosome biogenesis.

Even though the manuscript raises several interesting points, the study does not convincingly support some of the described relationships between p62 bodies and the autophagic machinery / membrane supply compartments. Although p62 bodies clearly interact with autophagic components, the specific relevance for the biogenesis is not sufficiently shown, especially with regard to p62 bodies being the starting point of autophagosome formation. How do the authors differentiate between membranes gathered to p62 bodies by p62 and not the other way around or being gathered there by other autophagic components in a secondary event? Can the authors clarify why they think autophagic components are recruited solely by p62 bodies and not by p62 in general? Here, the authors need to provide additional experiments. With these substantial concerns in mind, the following points need to be addressed:

- 1) For their MS-based profiling (Fig. 1), the authors mention that they optimized a previously published workflow. However, they do not describe what they adjusted. Also, to improve accessibility, the authors should describe all experiments in detail in the methods section.
- 2) The proteomic profiling of p62 bodies (Fig. 1) was performed with artificially constituted structures. Can the authors perform a similar proteomic profiling with cell-derived bodies, e.g. by employing APEX2-based proximity profiling? This would significantly enhance the data's physiological relevance and could also give insights into the positioning of these p62 bodies (e.g. in close proximity to specific ER regions).
- 3) When looking at GO terms enriched in the droplet fraction, no autophagy-specific terms were present. Why are these terms missing, especially when proposing a p62 body-specific recruitment? When analyzing the provided MS source table, ATG proteins do not seem to be enriched in the p62 body fraction compared to the controls. If sample columns were labelled correctly, this would actually point toward a decrease in the body fraction.
- 4) In extended Fig. 1d, the authors conclude proteome similarities based on correlation coefficients. Here, the authors should provide respective coefficients for all replicates. To dissect differences between the droplet fraction and the other fractions, a GO and GSEA comparison of those sample groups are needed (e.g. are vesicular terms also/similarly enriched in the controls?).
- 5) Instead of autophagic components, the MS screen proposes a variety of other vesicular compartments to be enriched at p62 bodies. The authors should check whether these components are actually recruited to p62 bodies or in reverse p62 bodies are targeted towards these structures? (e.g. colocalization analysis with markers for cellular compartments)
- 6) Regarding Fig. 2 colocalization: how did the authors segment between big and small bodies? More information about the quantification method is necessary. Additionally, the merged inlet for Fig. 2 ULK1 is not the same image as the single channels (intensity difference).
- 7) Since the colocalization in Fig. 2 a-b was only measured under starved conditions, is this colocalization increased compared to basal autophagy or other p62-inducing conditions?
- 8) In Fig. 2 c-d, the authors propose an altered colocalization of autophagic components with p62 bodies upon ATG2 deletion. To come to this conclusion, they compared the normalized intensity for singly, hand-selected dots. Here, a more unbiased quantification is necessary (e.g. measurement of intensities in p62-devoid areas, such as the rim of p62 bodies).
- 9) In Fig. 2 e, a co-stain with p62 is necessary to conclude that ATG9 and LC3 are accumulating around p62 bodies.
- 10) Some conclusions regarding the recruitment of membrane structures and autophagic components are based on ATG2 deletion (e.g. Fig. 2c-f and Fig. 3a-b). How do the authors exclude that these results are not just an (over-)compensation for dysfunctional autophagy? Can the authors show similar recruitment of ATG9-positive structures to p62 bodies upon functional

autophagy? If not, can the authors identify an isolation membrane or forming phagophore around p62 bodies in WT cells that is absent upon ATG2 depletion? The authors should use an autophagy-dysfunctional p62 variant that still forms p62 bodies as control.

11) In extended Fig. 3b-d the authors describe that p62 deletion had no effect on the capacity to form autophagosomes. However, in extended Fig. 3c a decrease in LC3 puncta is visible, please change this image to represent the quantification in extended Fig. 3d.

12) In Fig. 3 f, a decrease of expression levels of both, ATG9A and ATG16L1, was proposed under fed and starved conditions. However, the statistics only depict one test per protein. The authors should add the second comparison per protein. How is this decrease in total abundance correlating with the increase in visible structures in the immunofluorescence data?

13) The authors should perform quantification of the effects of Hexanediol (Fig. 3i). Again, it would be beneficial to show a similar effect (or with the forming phagophore as an alternative) in ATG2 WT cells.

14) For controlling the recruitment of membrane proteins to p62 bodies, the authors used PB1-deficient p62 in Fig. 4b. If PB1-mutant p62 is not able to reconstitute p62 bodies, how did the authors pull down p62-positive fractions and did the authors probe for p62 in respective fractions? If no p62 bodies are formed and thus no p62-positive fraction is enriched, no other protein (e.g. ATG9A) should be pulled down.

15) In extended Fig. 4c-d, colocalization of p62 with ATG16L1 and ATG9A is depicted for different conditions. How do the authors explain the observed increase in diffuse cytosolic staining for both proteins in PB1-mutant p62? Here, both stainings seem to follow the diffuse p62 staining, suggesting that even diffuse p62 might recruit ATG16L1 and ATG9A.

16) Regarding the recruitment of liposomes to p62 bodies (Fig. 4e-f), how did the authors ensure that this is p62 body specific and not a general mechanism by which (possibly hydrophobic) surfaces are engulfed by liposomes?

17) In Fig. 5 d the authors describe an ATP-dependent enrichment of PI3KC3 complex components. Especially regarding p-ATG14 enrichment, the authors need to quantify these effects, normalized to the efficiency of p62 body sedimentation. Similarly, results presented in Fig. 5 f need to be quantified in an unbiased manner. Also, did the authors use mCherry-p62 (axis description) or GFP-p62 (figure description)?

Reviewer #2 (Remarks to the Author):

This manuscript entitled "local membrane source gathering by p62 body drives autophagosome formation" by Xuezhao Feng et al. describes the role of p62 body in autophagosome formation especially in cargo selective autophagy. They showed that Atg9 and Atg16L1 is enriched around p62 body in ATG2 KO cells. They further demonstrated that p62 body serves as a platform to recruit PI3-kinase complex and phosphorylate them.

In general, this study is nicely showing critical role of p62 condensate in recruiting Atg proteins. Data are mostly clean and convincing. However, the following concerns should be clarified to be finally accepted.

1. In figure 2, percentage of each Atg positive p62 body in Atg2KO cells should be presented just like Atg9 and Atg16 in Figure 3b. If Atg16 is almost 100 % positive, LC3 should be expected either. Also, it is not clear to me why discrepancy exist in the value of Atg16 between figure 2b and 3b.

2. It is still incomprehensible why p62 body plays specifically in selective autophagy but not in nonselective autophagy. Figure E3e does not support the authors' claim, as p62 is essential for formation of ubiquitin condensate (Komatsu, Cell). What is strange is that in figure 3d, meanwhile Atg9 is accumulated in tubular structure, nonselective autophagy is still intact. What is that tubular structure? Do the other Atg proteins are recruited to there? Is there some rationalization such structure is formed in p62KO cells?

3. I am not assured how specific to atg proteins in this p62 body recruitment phenomenon. As the

authors mentioned, other membrane related proteins were also recruited in mass spec analysis. The authors need to evaluate the contribution of LC3 interacting motif of P62 in these processes.

Reviewer #3 (Remarks to the Author):

In this manuscript by Feng et al., the authors systematically identified the proteins isolated by p62 phase separated bodies, and proposed a mechanism model for p62 body function in autophagosome formation. The authors took an in vitro assay to purify p62 body bound proteins and then identified these proteins, followed by bioinformatics analysis. Based on the results that several autophagy proteins were found with p62 bodies, the authors confirmed that several autophagy proteins like ATG16L1, ULK1 and WIPI2 were nucleated by p62 bodies, which was not found before. More importantly, the authors found autophagy related membrane vesicles and associated proteins like ATG9A and ATG16L1 were also bound/nucleated by p62 bodies. The recruitment of ATG9A and ATG16L1 vesicles by p62 bodies were confirmed by in vivo and in vitro assays. Lastly, the authors found p62 bodies can spatially regulate PI3KC3-C1 kinase activities, leading to local PI3P generation and membrane tethering for autophagosome formation. The mass-spec identification of proteins in phase separated condensates/body is nice. In general, this study is interesting and systematically clarified the roles of p62 condensates/bodies in regulating autophagosome formation. In a big picture, this study also proposed a nice example to show the interplay between membrane-less condensates and membrane vesicles in regulating cellular functions, illustrating how the downstream signaling is transmitted by phase separated condensates/bodies.

There are several issues/points need to be addressed, as shown below.

1, Besides p62, the initiating complex Atg1-13-17 complex (in yeast) or FIP200 complex (in mammals) has also been found to exhibit phase separation, forming condensates/bodies. Can the authors analyze whether the phase separation of FIP200/ULK1 complex is disturbed in p62 knockout cells? Also, can the authors abolish both the p62 and the FIP200 phase separation, then analyze which one condensate is established first?

2, P62 body functions as a platform for recruiting several autophagy proteins, is there a time sequential for that recruitment? In other words, are these autophagy proteins recruited to p62 body one by one, or together simultaneously? This can be tested by experiments or discussed in the discussion section. As also mentioned by the authors, autophagosome formation requires a series of intricately regulated processes including core scaffold complex assembly, membrane precursor recruitment and modification, membrane expansion and shaping, and pore closure. Therefore, it is possible that the proteins function at different stages are recruited to p62 body sequentially.

3, How the p62 bodies specifically recruit these autophagy proteins? The authors also showed that p62 bodies recruit autophagy proteins like FIP200, ATG16L1, p-ATG16L1, ULK1, WIPI2, DFCP1 (Fig. 2a, b), while less or none colocalization was observed with ATG9A, ATG14 (Extended Data Fig. 2a, b). How this specificity is conferred? This can be discussed.

4, In Fig.1a, how the p62 droplet was isolated? The detailed information of this should be mentioned in methods section (centrifuge speed, time, washing?). Here, the droplet can be investigated by fluorescence to show that indeed the droplet contains p62 condensates/bodies.

5, In Fig. 1b, the bands between 40-50kDa in Mix and FT group but not in Cytosol group seem interesting. Have the authors analyzed what are they?

6, Several autophagy proteins like ATG2A/B, ATG3, ATG13, ATG101 and ATG16L1 were found within p62 droplet in the mass spectrometry data. ATG16L1 is confirmed. Can the authors confirm one or two more proteins from ATG2A/B, ATG3, ATG13, ATG101? If appropriate antibodies are available.

7, The surrounding of autophagy proteins outside of p62 body (Fig. 2c) by ATG2KO is interesting. Can the authors block the condensates (for example, FIP200 and p62) ATG2KO cells, to see whether they are still phase separated condensates?

8, In Extended Data Fig. 3a, cells were starved with DPBS (STA), and in Extended Data Fig. 3c, cells were starved with DMEM medium. It was planned for such difference in starvation conditions? The purpose of using STA should be mentioned.

9, In Fig4, the authors analyzed the proteins bound with the condensed liquid droplets. Can the authors also analyze the lipids bound with the condensed liquid droplets by Mass-spec, from WT

cytosol or ATG2KO cells?

10, What (1)-(3) in Fig.5a means? I guess these labeling was point to the quantification data in 5b. This should be mentioned in 5a legends.

11, There several places in the text need to be confirmed or corrected. (a), "The class III phosphatidylinositol 3-kinase complex I (PI3KC3-C1) (primarily containing VPS34, ATG14 and Beclin1) is activated at phagophore membrane....", here, should VPS15 be included in PI3KC3-C1? (b), "....PI3KC3-C1 complex and ULK1 in the presence or absence of ATP and analyzed the condensate or supernatant fraction by immunoblotting assay (Fig. 5c, b).", here, should "(Fig. 5c, b)" be "(Fig. 5c, d)"? (c), The title of Fig 5 legend, should "PI3K" be "PI3KC3-C1"?

Point-by-Point Response to the Reviewers' Comments

We would like to thank all the reviewers for their insightful comments and constructive suggestions. By performing new experiments and analyses, we have now thoroughly addressed each point of the concerns, and revised the manuscript accordingly.

The major efforts for the revision include:

- (1) perform APEX2-based proximity profiling coupled with mass spectrometry (MS) for *in vivo* identification of p62 adjacent proteins, as a complement to the original *in vitro* p62 droplet MS data.
- (2) validate a series of vesicle proteins in MS data for their colocalization with p62 bodies.
- (3) employ several p62 mutants with specific functional defects in the experiments to provide causal evidence for p62 body in recruiting membrane vesicles during autophagosome formation.
- (4) conduct a lipidomic profiling to identify the exact lipid components within p62 bodies which further substantiated our claims.
- (5) other experiments to address each point of concern.

The changes in the manuscript are marked **red**, and the point-by-point responses to the reviewers' comments are provided below.

Reviewer comments

Reviewer #1 (Remarks to the Author):

Autophagy, a complex mechanism for the degradation of a variety of cellular constituents, relies on the spatio-temporal assembly of specific components forming double-membraned structures called autophagosomes. During biogenesis and membrane elongation, the forming autophagosome relies on constant membrane supply. Detailed mechanisms behind this process, especially with regard to p62, a ubiquitous receptor for autophagic cargo, remain elusive. In the present manuscript Feng et al. describe membrane-recruitment, and potentially underlying mechanisms, to phase-separated p62 bodies. Upon profiling of p62 body composition by LC-MS, the authors describe the presence of autophagic components and membrane sources (e.g. ATG9 positive vesicles) in p62 bodies. This was enhanced in ATG2 deficient cells. Secondly, the authors propose that p62 bodies constitute a platform for the assembly of ULK1 and PI3KC3-C1 complexes, thus initiating autophagosome biogenesis.

Even though the manuscript raises several interesting points, the study does not convincingly support some of the described relationships between p62 bodies and the autophagic machinery / membrane supply compartments. Although p62 bodies clearly interact with autophagic components, the specific relevance for the biogenesis is not sufficiently shown, especially with regard to p62 bodies being the starting point of autophagosome formation. How do the authors differentiate between membranes gathered to p62 bodies by p62 and not the other way around or being gathered there by other autophagic components in a secondary event? Can the authors clarify why they think autophagic components are recruited solely by p62

bodies and not by p62 in general? Here, the authors need to provide additional experiments. With these substantial concerns in mind, the following points need to be addressed:

Response: We appreciate the reviewer for the positive comments and insightful criticisms of our work. We have performed a series of new experiments to address these concerns and revised the manuscript accordingly. To further link p62 bodies to the biogenesis of autophagosome, we performed APEX2-based proximity profiling for p62 adjacent proteins as an *in vivo* supplement to p62 droplet mass spectrometry and again identified autophagic machinery and membrane components around p62. To determine if membranes are gathered to p62 bodies or the other way around or by p62 in general, we employed multiple p62 mutants to examine their capabilities to recruit membranes in the revision. Moreover, we conducted a lipidomic profiling to identify the exact lipid components within p62 bodies which further strengthen our original discovery. We also performed other experiments to address every concern from the reviewer. The results and responses are elaborated as follows.

1) For their MS-based profiling (Fig. 1), the authors mention that they optimized a previously published workflow. However, they do not describe what they adjusted. Also, to improve accessibility, the authors should describe all experiments in detail in the methods section.

Response: We had added the detailed information of phase separation, sample preparation and bioinformatics analysis in the **Methods** section.

2) The proteomic profiling of p62 bodies (Fig. 1) was performed with artificially constituted structures. Can the authors perform a similar proteomic profiling with cell-derived bodies, e.g. by employing APEX2-based proximity profiling? This would significantly enhance the data's physiological relevance and could also give insights into the positioning of these p62 bodies (e.g. in close proximity to specific ER regions).

Response: Using NRK cells expressing APEX2-GFP-p62, we conducted proximity profiling experiment to identify *in vivo* protein environment of p62. Effective expression of APEX2-GFP-p62 and proximity labeling of biotinylation signal were achieved for three biological replicates (**Supplementary Fig. 2a-c**). After streptavidin pull-down, biotinylated proteins were identified with mass spectrometry (**Methods**). We identified 241 proteins that are specifically enriched around p62 protein (**Supplementary Table 4**). Consistently, a majority of hits (157 out of 241) were also present in the p62 droplet fraction identified via *in vitro* phase separation (**Supplementary Fig. 2e**), indicating a high agreement between the two approaches. Notably, compared to *in vitro* condition, not all the p62 forms droplet in proximity profiling condition. Thus, this observed overlap might still be underestimated. Interestingly, among the p62 adjacent hits by proximity profiling, the top significantly enriched functional terms included “vesicle-mediated transport”, “membrane trafficking” and “selective autophagy”, again showing the physiological relevance of p62 and consistence with those *in vitro* data. These new *in vivo* data complement the *in vitro* p62 droplet proteomics profiling results and further consolidate the physiological relevance of p62 bodies with vesicle components and autophagy.

Supplementary Fig. 2 APEX2-based proximity profiling of p62 adjacent proteins.

a, Western blot analysis after APEX2-p62 activation and streptavidin pull-down of biotinylated proteins. Three biological replicates were performed. The control group was treated in parallel but without H₂O₂ stimulus compared to label group.

b, SDS PAGE separation of the control and label samples and digested by trypsin. Each sample was cut into one fraction and analyzed by LC-MS/MS.

c, Number of proteins identified and median intensity of the control (blue bar) and label (orange bar) experiment.

d, Dot plot showing the APEX-enriched p62 adjacent proteins (highlighted in red dots) over the control.

e, Venn diagram showing the overlap between APEX-enriched p62 adjacent proteins and p62 droplet proteins.

f, Metascape enrichment of the proteins enriched in the APEX labeling group (n=241). The top 20 enriched main cluster terms of biological pathways were representing. The sub clusters of “*Salmonella* infection” (purple) and “Selective autophagy” (orange) were shown.

3) When looking at GO terms enriched in the droplet fraction, no autophagy-specific terms were present. Why are these terms missing, especially when proposing a p62 body-specific recruitment? When analyzing the provided MS source table, ATG proteins do not seem to be enriched in the p62 body fraction compared to the controls. If sample columns were labelled correctly, this would actually point toward a decrease in the body fraction.

Response: We re-analyzed the 590 enriched proteins in the p62 droplet (**Supplementary Table 2**) for functional term enrichment analysis through Metascape (<https://metascape.org/gp/index.html#/main/step1>), and the “autophagy” term was present as the 10th-ranked biological processes (**Fig. 1f; Supplementary Table 3**). Several autophagy machinery proteins are enriched, like NBR1, STX17 and GABARAP. Although some ATG proteins were not “enriched” in p62 body fraction, they were indeed present in this droplet fraction (**Supplementary Table 2**). We argue that a protein has not to be “enriched” in p62 bodies to play important function because that : 1) the relative enrichment can be biased by the association strength of a given protein to p62 bodies with stronger interaction showing higher enrichment and vice versa; 2) proteins with weak interaction and less enrichment in p62 bodies may still exert important functions (**Supplementary Table 2**).

Fig. 1f, Metascape enrichment of proteins enriched in the droplet (n = 590). The top 20 enriched main cluster terms of biological pathways were presented.

4) In extended Fig. 1d, the authors conclude proteome similarities based on correlation coefficients. Here, the authors should provide respective coefficients for all replicates. To dissect differences between the droplet fraction and the other fractions, a GO and GSEA comparison of those sample groups are needed (e.g. are vesicular terms also/similarly enriched in the controls?).

Response: Following the reviewer's suggestion, we added the Pearson correlation coefficients for all the replicates between the Mix, FT and droplet samples in the revised version (**Supplementary Fig. 1c**). The coefficients were high between droplets and low with FT or Mix. Moreover, we added the GO and GSEA analysis of proteins enriched for both droplet and Mix groups in **Supplementary Fig. 1d-g**. The vesicular terms were significantly enriched in the droplet but not in the control mixture.

Supplementary Fig. 1c. Pearson correlation coefficients of three biological replicates between Mix, FT and droplet.

Supplementary Fig. 1d, Gene Set Enrichment Analysis (GSEA) for GO biological processes (BP) of proteins enriched in droplet (up, orange panel) and Mix (down, blue panel).

e, Enrichment plots for the top two data set enriched in GSEA BP analysis. Gene sets of “Regulation of exocytosis” was significantly enriched in droplet.

f, GSEA for GO cellular components (CC) of proteins enriched in droplet (up, orange panel) and Mix (down, blue panel).

g, Enrichment plot for the top data set enriched in GSEA CC analysis. Gene set of “SNARE complex” was significantly enriched in droplet.

5) Instead of autophagic components, the MS screen proposes a variety of other vesicular compartments to be enriched at p62 bodies. The authors should check whether these components are actually recruited to p62 bodies or in reverse p62 bodies are targeted towards these structures? (e.g. colocalization analysis with markers for cellular compartments)

Response: Using immunofluorescence assay, we checked the colocalization of several other MS-identified vesicle proteins with p62 bodies. We found that SNARE components (STX17, STX12, SNX2, VAMP3, STX6 and STX3) and several RAB family members (RAB2A, RAB1A, RAB5B and RAB5C) were indeed colocalized with p62 bodies (**Supplementary Fig. 3a, b**), confirming the association of these diverse vesicle components to p62 bodies.

a

b

Supplementary Fig. 3 Validation of colocalization of membrane vesicle proteins and ATG proteins with p62 bodies.

a, EGFP-tagged RAB5B, RAB5C, RAB1A, RAB2A, STX17, STX6, STX12 and STX3 were transiently expressed in NRK cells, starved for 4h, and cells were fixed and stained with antibodies against GFP and p62. Scale bar, 10 μ m.

b, EGFP-tagged ATG2A, ATG2B were transiently expressed in NRK cells. The cells were starved for 4 h, then stained with antibodies against GFP and p62 (upper panels). WT cells starved for 4 h were stained with antibodies against p62 and ATG3 or ATG13. Scale bar, 10 μ m.

6) Regarding Fig. 2 colocalization: how did the authors segment between big and small bodies? More information about the quantification method is necessary.

Response: Big p62 bodies were defined as those with a diameter larger than 300 nm while the diameter of small ones was under 300 nm. The quantification was performed by IMAGE J. We have added this information in the **Methods** section of the revised manuscript.

7) Since the colocalization in Fig. 2 a-b was only measured under starved conditions, is this colocalization increased compared to basal autophagy or other p62-inducing conditions?

Response: In normal medium with basal autophagy, the formation of p62 bodies was significantly reduced compared to starved conditions with high level of autophagy. Accordingly, the colocalization of autophagic proteins with p62 was less evident (**Reviewer only Fig. 1**). We also examined cells with p62 overexpression and the colocalization was significantly increased compared to basal autophagy (**Reviewer only Fig. 2**).

Reviewer only Fig. 1 The colocalization of autophagic proteins with p62 bodies in basal autophagy.

a, Cells were stained with antibodies against p62 and FIP200, p-ATG16L1, WIPI2, ATG9A. Scale bar, 10 μ m.

b, The number of p62 bodies were quantified in cells from a (3 independent experiments; 50 cells were assessed per independent experiment).

c, The percentage of indicated protein-positive p62 bodies were quantified from data in (a) (3 independent experiments; 100 puncta were assessed per independent experiment).

Reviewer only Fig. 2 The colocalization of autophagic proteins with p62 bodies in p62 overexpression cells.

a, EGFP-tagged p62 was transiently expressed in NRK cells. Cells were treated with rapamycin for 2 h, then stained with antibodies against GFP and FIP200, p-ATG16L1, WIPI2 or ATG9A. The left panel shows the co-localization of FIP200, p-ATG16L1, WIPI2 or ATG9A with p62. Right panels show enlarged p62 structures that were either FIP200, p-ATG16L1, WIPI2 or ATG9A positive (upper panels) or FIP200, p-ATG16L1, WIPI2 and ATG9a negative (lower panels) (Big p62 bodies is \geq 300 nm and small were under 300 nm). Scale bar, 10 μ m.

b, The percentage of FIP200, p-ATG16L1 and WIPI2 positive puncta in micron-scale (Micron) or nanoscale (Nano) p62 bodies quantified in cells from a (3 independent experiments; 100 puncta were assessed per independent experiment). P values were calculated using the two-tailed, unpaired t-test. * $P < 0.05$, ** $P < 0.01$, *** $P < 0.001$.

c, The percentage of indicated protein-positive p62 bodies were quantified from data in (a) (3 independent experiments; 100 puncta were assessed per independent experiment).

8) In Fig. 2 c-d, the authors propose an altered colocalization of autophagic components with p62 bodies upon ATG2 deletion. To come to this conclusion, they compared the normalized intensity for singly, hand-selected dots. Here, a more

unbiased quantification is necessary (e.g. measurement of intensities in p62-devoid areas, such as the rim of p62 bodies).

Response: Following the reviewer's suggestion, we used a more unbiased quantification approach by simultaneously measuring the intensities in the rim of p62 bodies in this analysis (**Fig. 2c, d**). The original conclusion was better supported by the revised figure.

Fig. 2c, *Atg2ab* DKO and NRK cells were starved for 4 h, and then stained with antibodies against FIP200, WIPI2, LC3, ATG14 and p62. Scale bars, 10 μm.

d, Line profiling of a representative section of the cell, indicated by the blue lines in **c**.

9) In Fig. 2 e, a co-stain with p62 is necessary to conclude that ATG9 and LC3 are accumulating around p62 bodies.

Response: We co-stained p62 with ATG9A and LC3 in new **Fig. 2e** as the reviewer suggested. Significant colocalization of ATG9A and LC3 with p62 bodies was observed.

Fig. 2e, *Atg2ab* DKO cells were stained with antibodies against p62, ATG9A and LC3. Scale bars, 5 μ m.

10) Some conclusions regarding the recruitment of membrane structures and autophagic components are based on ATG2 deletion (e.g. Fig. 2c-f and Fig. 3a-b). How do the authors exclude that these results are not just an (over-)compensation for dysfunctional autophagy? Can the authors show similar recruitment of ATG9-positive structures to p62 bodies upon functional autophagy? If not, can the authors identify an isolation membrane or forming phagophore around p62 bodies in WT cells that is absent upon ATG2 depletion? The authors should use an autophagy-dysfunctional p62 variant that still forms p62 bodies as control.

Response: In WT cells with functional autophagy, the recruitment of ATG9A-positive structures to p62 bodies can be observed although to a significantly lesser extent compared to that in *Atg2ab* DKO cells (**Fig. 3b**). This is possibly because the recruitment and merge of ATG9A-vesicles into larger isolation membrane is a concomitantly fast process. Thus, it is hard to snap the transient recruitment state unless blocking the membrane fusion by knocking out ATG2. Using trans electron microscopy (TEM), we did observe the isolation membrane around p62 bodies in WT cells (**Reviewer only Fig. 3**). Moreover, we further employed a p62 mutant (p62-LIR mut) that disrupts functional LC3 interacting region and examined its capability to recruit ATG9A and ATG16L1 vesicles as well as other ATG proteins. we observed that this mutant could still form droplet and was also capable to recruit ATG9A and ATG16L1 vesicles as well as core autophagosome machineries, although to a lesser extent to that of WT p62 (**Supplementary Fig. 8**). In contrast, another p62 mutant (p62 M404V) that fails to be phase separated did not recruit ATG9A and ATG16L1 vesicles in either *Atg2ab* DKO cells or *p62* KO cells (**Supplementary Fig. 7**). These data indicate that it is the p62 body but not diffused form of p62 that can recruit membrane vesicles.

Reviewer only Fig. 3 Isolation membrane around p62 bodies in WT cells by trans electron microscopy. TEM image showing the DAB staining pattern in WT cells transiently transfected with GFP-APEX2-p62. Scale bar, 500nm, 200nm.

Supplementary Fig. 8 The recruitment of membrane vesicles and autophagy proteins by p62-LIR mutant.

a, EGFP-tagged p62 or p62-LIR mutant were transiently expressed in NRK cells, and cells were stained with antibodies against GFP and FIP1200, p-ATG16L1, ATG9A, WIPI2 or LC3. Scale bar, 10 μ m.

b, The percentage of indicated protein-positive p62 bodies were quantified from data in (a) (3 independent experiments; 100 puncta were assessed per independent experiment). The P value was calculated using the two-tailed, unpaired t -test. **** $P < 0.0001$, ns means not significant.

Supplementary Fig. 7 The recruitment of membrane vesicles and autophagy proteins by phase separation deficient p62-M404V mutant.

a-b, EGFP-tagged p62 or p62-M404V was transiently expressed in *Atg2ab* DKO cells (**a**) and *p62* KO cells (**b**). The cells were then stained with antibodies against GFP, ATG9A, FIP200, p-ATG16L1 and WIPI2. Scale bar, 20 μ m.

11) In extended Fig. 3b-d the authors describe that p62 deletion had no effect on the capacity to form autophagosomes. However, in extended Fig. 3c a decrease in LC3 puncta is visible, please change this image to represent the quantification in extended Fig. 3d.

Response: We thank the reviewer for this suggestion and have changed the image (new **Supplementary Fig. 5c**) to better present the results.

Supplementary Fig. 5c, WT and *p62* KO cells were starved with DMEM medium for 4 h, and then stained with antibodies against LC3 or FIP200. Scale bar, 10 μ m.

12) In Fig. 3 f, a decrease of expression levels of both, ATG9A and ATG16L1, was proposed under fed and starved conditions. However, the statistics only depict one

test per protein. The authors should add the second comparison per protein. How is this decrease in total abundance correlating with the increase in visible structures in the immunofluorescence data?

Response: We performed three independent tests for this Western blot analysis (raw data provided in (Reviewer only Fig. 4) and quantified data in new Fig. 3f were based on these three tests. ATG9A was significantly reduced in *p62* KO cells while ATG16L1 only slightly dropped upon *p62* knockout (Fig. 3f).

Reviewer only Fig. 4 Raw images for three independent tests of Western blot for Fig. 3f NRK cells or *p62* KO cells were starved with DPBS (STA) for 4 h, and then the cell lysates were analyzed by immunoblotting with antibodies against ATG9A, pATG16L1 and GAPDH.

Fig.3f The ratio of ATG9A or pATG16L1 to GAPDH were analyzed from **e**. *P* values were calculated using the two-tailed, unpaired t-test, *n*=3, **P*<0.05.

The decrease in total abundance of ATG9A in *p62* KO cells indicates a direct and/or indirect role of *p62* in regulating total levels of these two proteins. Meanwhile, loss of *p62* significantly altered the cytoplasmic localization of ATG9A and p-ATG16L1 by concentrating these proteins around a perinuclear structure despite a decrease of

total ATG9A. Local increase of protein concentration here was due to altered spatial distribution but not total abundance of the protein.

13) The authors should perform quantification of the effects of Hexanediol (Fig. 3i). Again, it would be beneficial to show a similar effect (or with the forming phagophore as an alternative) in ATG2 WT cells.

Response: We performed quantification analysis for original **Fig. 3i** and the results were shown in new **Supplementary Fig. 9a**. We also conducted similar assays for FIP200 (**Supplementary Fig. 9b**). It seemed that p62 bodies reformed at first, then other components such as FIP200, ATG9A and ATG16L1 vesicles started to concentrate around p62 bodies. These data support p62 as a spatial organizer to assemble core autophagy machinery. As the colocalization of ATG9A with p62 body is not that dramatic in WT cells, it is technically challenging to count enough signals for quantification analysis. Instead, we did observe forming phagophore around p62 body in WT cells by TEM (**Reviewer only Fig. 3**).

Supplementary Fig. 9 The recovery of p62 bodies after 1,6-hexanediol removal.

a, The number of indicated protein-positive puncta was quantified in Fig. 3i (3 independent experiments; 100 puncta were assessed per independent experiment).

b, *Atg2ab* DKO cells were starved for 2 h and cells were treated with 2% 1,6-hexanediol for 20 min (H). The cells were then recovered by washing with PBS and incubation with complete medium (HR) for 3min, 5min, 10min or 15min. Then cells were stained with an antibody against p62 and FIP200. Scale bar, 10 μ m.

14) For controlling the recruitment of membrane proteins to p62 bodies, the authors used PB1-deficient p62 in Fig. 4b. If PB1-mutant p62 is not able to reconstitute p62

bodies, how did the authors pull down p62-positive fractions and did the authors probe for p62 in respective fractions? If no p62 bodies are formed and thus no p62-positive fraction is enriched, no other protein (e.g. ATG9A) should be pulled down.

Response: We employed PB1-deleted p62 which is not able to reconstitute p62 bodies as a negative control for membrane protein recruitment. Indeed, as shown in the lane 7 of **Fig. 4b**, the pellet fraction of PB1-mutant p62 group did not contain p62 nor other proteins tested, suggesting that p62 body formation is a prerequisite for membrane protein recruitment. Moreover, we have added a new phase separation-deficient p62 mutant (p62 M404V) that fails to be phase separated in *Atg2ab* DKO cells and found that such mutant did not recruit ATG9A and ATG16L1 vesicles as well as FIP200 and WIPI2 puncta (**Supplementary Fig. 7a**). Similar findings were also shown in p62 KO cells (**Supplementary Fig. 7b**). These data indicate that it is the p62 body but not diffused form of p62 that can recruit membrane vesicles.

Supplementary Fig. 7 The recruitment of membrane vesicles and autophagy proteins by phase separation deficient p62-M404V mutant.

a-b, EGFP-tagged p62 or p62-M404V was transiently expressed in *Atg2ab* DKO cells (**a**) and p62 KO cells (**b**). The cells were then stained with antibodies against GFP, ATG9A, FIP200, p-ATG16L1 and WIPI2. Scale bar, 20 μ m.

15) In extended Fig. 4c-d, colocalization of p62 with ATG16L1 and ATG9A is depicted for different conditions. How do the authors explain the observed increase in diffuse cytosolic staining for both proteins in PB1-mutant p62? Here, both stainings seem to follow the diffuse p62 staining, suggesting that even diffuse p62 might recruit ATG16L1 and ATG9A.

Response: We double checked the image here and found that the increased diffuse signals in PB1-mutant p62 group were largely due to higher background of the whole image. We have changed the corresponding images with uniform background in new **Supplementary Fig. 10a-d**.

Supplementary Fig. 10 Recruitment of lipid membrane by p62 bodies *in vitro*.

a-d, p62 body and different protein /protein -interacting vesicles from Fig. 4c a were co-stained with antibodies against FIP200, ATG9A, ATG16L1 and LC3. Scale bar, 2 μm.

16) Regarding the recruitment of liposomes to p62 bodies (Fig. 4e-f), how did the authors ensure that this is p62 body specific and not a general mechanism by which (possibly hydrophobic) surfaces are engulfed by liposomes?

Response: To determine whether the recruitment of liposomes to p62 bodies is specific, we employed several well-characterized and independent phase-separated protein condensates such as SUMO, FUS, DDX4 and HURNPAB, and examined their abilities to attract liposomes. As shown in **Supplementary Fig. 11a, b**, compared to p62 bodies, all the other four protein concentrates had significantly lower capabilities to recruit liposomes. These data go against the general model of

hydrophobic surface engulfment by liposomes and support a p62 body-specific mechanism in recruiting liposomes.

Supplementary Fig. 11 The recruitment of liposomes by different phase separated condensates.

a, Liposomes (PE-NDB) around the phase separated p62, SUMO, FUS, DDX4 and HURNPAB droplets were observed. Scale bar, 1 μ m.

b, The percentage of p62 droplet colocalization with liposome was quantified in images from (a) (3 independent experiments; 100 puncta were assessed per experiment). The *P* value was calculated using the two-tailed, unpaired t-test.

17) In Fig. 5 d the authors describe an ATP-dependent enrichment of PI3KC3 complex components. Especially regarding p-ATG14 enrichment, the authors need to quantify these effects, normalized to the efficiency of p62 body sedimentation. Similarly, results presented in Fig. 5 f need to be quantified in an unbiased manner. Also, did the authors use mCherry-p62 (axis description) or GFP-p62 (figure description)?

Response: We thank the reviewer for this suggestion. As a more appropriate approach, we quantified p-Beclin1 and p-ATG14 by normalizing to the corresponding total Beclin1 or ATG14 in each sample. We also presented quantified data for original Fig. 5f in new **Fig. 5h**. Furthermore, we used GFP-p62 in original Fig. 5f and

mislabeled as mCherry-p62 in axis description. This typo has been corrected in new Fig. 5g, h.

Fig. 5d, The sedimentation and supernatant were separated by centrifugation and analyzed by western blot using antibodies against p-Beclin1, Beclin1, p-ATG14, ATG14 and ULK1. S: supernatant, P: pellet.

e, The ratio of p-Beclin1 or p-ATG14 to total Beclin1 or ATG14 were analyzed from **d**, *P* values were calculated using the two-tailed, unpaired t-test, *n*=3, *****P*<0.0001.

Fig. 5g, Liposomes (PE-NDB) and mCherry-FYVE(X2) around the phase separated p62 bodies were observed from **c**. Scale bar, 1 μ m.

h, The number of mCherry-FYVE puncta colocalized with p62 droplet was quantified. (*n* = 3 independent experiments. w/o means without. *P* values were calculated using the two-tailed, unpaired t-test. **** *P*<0.0001.

Reviewer #2 (Remarks to the Author):

This manuscript entitled “local membrane source gathering by p62 body drives autophagosome formation” by Xuezhao Feng et al. describes the role of p62 body in autophagosome formation especially in cargo selective autophagy. They showed that Atg9 and Atg16L1 is enriched around p62 body in ATG2 KO cells. They further demonstrated that p62 body serves as a platform to recruit PI3-kinase complex and phosphorylate them.

In general, this study is nicely showing critical role of p62 condensate in recruiting Atg proteins. Data are mostly clean and convincing. However, the following concerns should be clarified to be finally accepted.

Response: We are very grateful to the reviewer for the positive comments of our work. Meanwhile, we have thoroughly addressed the concerns from the reviewer by performing new experiments and improved the manuscript accordingly.

1. In figure 2, percentage of each Atg positive p62 body in Atg2KO cells should be presented just like Atg9 and Atg16 in Figure 3b. If Atg16 is almost 100 % positive, LC3 should be expected either. Also, it is not clear to me why discrepancy exist in the value of Atg16 between figure 2b and 3b.

Response: Following the reviewer’s suggestion, we have added the new quantification data of various autophagy proteins in Fig. 2 as the way presented in Fig. 3b (new **Supplementary Fig. 4e**). Consistently, all these tested proteins exhibited significantly higher colocalization with p62 bodies in *Atg2ab* DKO cells than that in WT cells. We also examined LC3 which indeed showed almost 100% colocalization with p62 bodies (**Supplementary Fig. 4e**). The difference of Atg16 value in Fig. 2b and Fig. 3b lies in that the latter one showing the high percentage of colocalization was done in *Atg2ab* DKO cells whereas in the former WT cells with only basal level of autophagy the degree of colocalization was lower.

Supplementary Fig. 4e. The percentage of indicated protein-positive p62 bodies in either WT and *Atg2ab* DKO cells. (3 independent experiments; 100 puncta were assessed per independent experiment). *P* values were calculated using the two-tailed, unpaired t-test, ***P*<0.01 *****P*<0.0001.

2. It is still incomprehensible why p62 body plays specifically in selective autophagy but not in nonselective autophagy. Figure E3e does not support the authors' claim, as p62 is essential for formation of ubiquitin condensate (Komatsu, Cell). What is strange is that in figure 3d, meanwhile Atg9 is accumulated in tubular structure, nonselective autophagy is still intact. What is that tubular structure? Do the other Atg proteins are recruited to there? Is there some rationalization such structure is formed in p62KO cells?

Response: Accumulating evidence, together with our results in p62 KO cells, support a more critical role of p62 bodies in selective autophagy than that in starvation-induced non-selective autophagy^{1, 2}. Our data showed that, in p62 KO cells, the number of LC3 and FIP200 puncta only slightly decreased (**Supplementary Fig. 5c, d**). This is consistent with the previous reports^{1, 2}. For example, in the Cell paper (Komatsu, Cell, 2007)², the authors generated p62 knockout mice and claimed that “The conversion from LC3-I to LC3-II, induction of GFP-LC3 dots, and the appearance of many autophagosome structures after starvation were similar between the control and p62-deficient hepatocytes (Figures S9A, S9B, and S9C).” This phenomenon is probably because that: 1) other signals such as TORC1 inactivation trigger Atg1/ULK assemblage formation and kinase activation to initiate non-selective autophagy, whereas selective autophagy depends on specific cargo receptor such as p62 to trigger autophagosome formation; (2) multiple cargo receptors may functionally compensate each other in some scenarios; and (3) the action mode of autophagy significantly differs in different tissues and/or cell states, which is difficult to make a general judgement for p62 essentiality in selective or non-selective autophagy.

Supplementary Fig. 5c, WT and p62 KO cells were starved with DMEM medium for 4 h, and then stained with antibodies against LC3 or FIP200. Scale bars, 10µm.

d, The number of LC3 or FIP200 puncta was quantified in images from **c** (3 independent experiments; 50 cells were assessed per independent experiment). The *P* value was calculated using the two-tailed, unpaired t-test.

In original Figure E3e, we showed that ubiquitinated proteins (but not ubiquitin condensates) are increased in p62 KO cells, possibly due to reduced conjugation to p62 and consequent failure of degradation via p62-Ub conjugation systems. This is consistent with data shown in Komatsu paper (Komatsu, Cell, 2007) that p62 is essential for the formation of ubiquitin condensate. Those ubiquitinated proteins failed to form condensate with p62 showed decreased p62-mediated degradation, thereby leading to increased accumulation.

To determine the identity of the tabular structure, we tested several organelle markers including sec61b for ER, GM130 for Golgi and ERGIC53 for ERGIC. We found that Golgi marker GM130 was significantly co-stained with ATG9A in p62 KO cells (**Supplementary Fig. 6a**), indicating that such tabular structure is from Golgi complex. Other ATG proteins such as FIP200, DFPC-1, ATG16, ATG14, WIPI-2 and LC3 were not recruited to that structure in p62 KO cells (**Supplementary Fig. 6b**), suggesting that, without p62 bodies, ATG9A vesicles cannot be properly recruited to growing autophagosome. These data also indicate that Golgi might be a primary source or relay station of ATG9A vesicles and without p62 body guidance these vesicles tend to be stuck in such position.

Supplementary Fig. 6 Aberrant localization of ATG9 vesicles and other ATG proteins in p62 KO cells.

a, EGFP-tagged SEC61 β or GFP-ERGIC53 was transiently expressed in p62 KO cells. The cells were stained with antibodies against GFP and ATG9A. p62 KO cells were stained with antibodies against GM130 and ATG9A. Scale bar, 10 μ m.

b, EGFP-tagged FIP200, DFPC1, ATG16, ATG14 or WIPI2 were transiently expressed in p62 KO cells, and the cells were stained with antibodies against GFP and ATG9A. p62 KO cells were starved and stained with antibodies against LC3 and ATG9A. Scale bar, 10 μ m.

3. I am not assured how specific to atg proteins in this p62 body recruitment phenomenon. As the authors mentioned, other membrane related proteins were also recruited in mass spec analysis. The authors need to evaluate the contribution of LC3 interacting motif of P62 in these processes.

Response: The recruitment of membrane related proteins to p62 bodies is specific. Using immunofluorescence assay, we further validated the co-localization of several ATG proteins with p62 bodies, including ATG2A/2B, ATG3, ATG13 and ATG101 (**Supplementary Fig. 3b**). In addition, we also validated other vesicle related proteins in mass spec analysis and found that SNARE components (STX17, STX12, SNX2, VAMP3, STX6 and STX3) and several RAB family members (RAB2A, RAB1A, RAB5B and RAB5C) were also colocalized with p62 bodies (**Supplementary Fig. 3a**). In contrast, other RAB proteins which were not identified by MS such as RAB29 was not (**reviewer only Fig. 5**).

Supplementary Fig. 3 Validation of colocalization of membrane vesicle proteins and ATG proteins with p62 bodies.

a, EGFP-tagged RAB5B, RAB5C, RAB1A, RAB2A, STX17, STX6, STX12 and STX3 were transiently expressed in NRK cells, starved for 4h, and cells were fixed and stained with antibodies against GFP and p62. Scale bar, 10 μ m.

b, EGFP-tagged ATG2A, ATG2B were transiently expressed in NRK cells. The cells were starved for 4 h, then stained with antibodies against GFP and p62 (upper panels). WT cells starved for 4 h were stained with antibodies against p62 and ATG3 or ATG13. Scale bar, 10 μ m.

Reviewer only Fig. 5 RAB29 does not colocalize with p62 bodies. EGFP-tagged RAB29 was transiently expressed in NRK cells. The cells were stained with antibodies against GFP and p62. Scale bar, 10 μ m.

As the reviewer suggested, we also employed a p62 mutant (p62-LIR mut) that disrupts functional LC3 interacting region and examined its capability to recruit ATG proteins. We found that this p62 mutant could still recruit ATG proteins like FIP200, ATG16 and WIPI2. As expected, p62-LIR mut did not recruit LC3 while WT p62 could (**Supplementary Fig. 8a, b**). These data suggest that p62 body could still recruit ATG proteins without direct LC3 interaction.

Supplementary Fig. 8 The recruitment of membrane vesicles and autophagy proteins by p62-LIR mutant.

a, EGFP-tagged p62 or p62-LIR mutant were transiently expressed in NRK cells, and cells were stained with antibodies against GFP and FIP200, p-ATG16L1, ATG9A, WIPI2 or LC3. Scale bar, 10 μ m.

b, The percentage of indicated protein-positive p62 bodies were quantified from data in (**a**) (3 independent experiments; 100 puncta were assessed per independent experiment). The *P* value was calculated using the two-tailed, unpaired t-test. *****P* < 0.0001, ns means not significant.

Reviewer #3 (Remarks to the Author):

In this manuscript by Feng et al., the authors systematically identified the proteins isolated by p62 phase separated bodies, and proposed a mechanism model for p62 body function in autophagosome formation. The authors took an in vitro assay to purify p62 body bound proteins and then identified these proteins, followed by bioinformatics analysis. Based on the results that several autophagy proteins were found with p62 bodies, the authors confirmed that several autophagy proteins like ATG16L1, ULK1 and WIPI2 were nucleated by p62 bodies, which was not found before. More importantly, the authors found autophagy related membrane vesicles and associated proteins like ATG9A and ATG16L1 were also bound/nucleated by p62 bodies. The recruitment of ATG9A and ATG16L1 vesicles by p62 bodies were confirmed by in vivo and in vitro assays. Lastly, the authors found p62 bodies can spatially regulate PI3KC3-C1 kinase activities, leading to local PI3P generation and membrane tethering for autophagosome formation.

The mass-spec identification of proteins in phase separated condensates/body is nice. In general, this study is interesting and systematically clarified the roles of p62 condensates/bodies in regulating autophagosome formation. In a big picture, this study also proposed a nice example to show the interplay between membrane-less condensates and membrane vesicles in regulating cellular functions, illustrating how the downstream signaling is transmitted by phase separated condensates/bodies.

There are several issues/points need to be addressed, as shown below.

Response: We appreciate the reviewer for the high recognition of the quality and significance of our work as well as those constructive comments and suggestions to improve the manuscript.

1, Besides p62, the initiating complex Atg1-13-17 complex (in yeast) or FIP200 complex (in mammals) has also been found to exhibit phase separation, forming condensates/bodies. Can the authors analyze whether the phase separation of FIP200/ULK1 complex is disturbed in p62 knockout cells? Also, can the authors abolish both the p62 and the FIP200 phase separation, then analyze which one condensate is established first?

Response: In starved p62 KO cells, the phase separation of FIP200 were significantly reduced (**Supplementary Fig. 5c, d**), further supporting a critical role of p62 bodies in the biogenesis of autophagosome. However, the number of FIP200 puncta in p62 KO cells was still maintaining at a relative high level, consistent with the notion that p62 knockout does not significantly affect non-selective autophagy^{1,2} (**Supplementary Fig. 5c, d**).

Supplementary Fig.5c, WT and *p62* KO cells were starved with DMEM medium for 4 h, and then stained with antibodies against LC3 or FIP200. Scale bars, 10 μ m.

d, The number of LC3 or FIP200 puncta was quantified in images from **c** (3 independent experiments; 50 cells were assessed per independent experiment). The *P* value was calculated using the two-tailed, unpaired t-test.

To determine which condensate is established first, we employed 1,6-hexanediol to firstly destroy both *p62* and FIP200 phase separation and examined their recovery rate after 1,6-hexanediol removal. As shown in **Supplementary Fig. 9b**, *p62* body and FIP200 condensate started to reform very quickly even after 3min of 1,6-hexanediol removal. With longer recovery times (10-15min), the FIP200 condensate apparently emerged surrounding *p62* bodies, suggesting that *p62* body may serve as an nucleation center to establish FIP200 puncta spatially.

b

Supplementary Fig. 9b, *Atg2ab* DKO cells were starved for 2 h and cells were treated with 2% 1,6-hexanediol for 20 min (H). The cells were then recovered by washing with PBS and incubation with complete medium (HR) for 3min, 5min, 10min or 15min. Then cells were stained with an antibody against p62 and FIP200. Scale bar, 10 μ m.

2, P62 body functions as a platform for recruiting several autophagy proteins, is there a time sequential for that recruitment? In other words, are these autophagy proteins recruited to p62 body one by one, or together simultaneously? This can be tested by experiments or discussed in the discussion section. As also mentioned by the authors, autophagosome formation requires a series of intricately regulated processes including core scaffold complex assembly, membrane precursor recruitment and modification, membrane expansion and shaping, and pore closure. Therefore, it is possible that the proteins function at different stages are recruited to p62 body sequentially.

Response: To examine the sequential recruitment of autophagy proteins to p62 bodies, we performed a time course experiment during p62 body reconstitution after 1,6-hexanediol removal. As shown in **Fig. 3i** and **Supplementary Fig. 9**, p62 bodies reformed at first, then other components such as FIP200, ATG9A and ATG16L1 vesicles started to concentrate around p62 bodies. These data support p62 as a spatial organizer to assemble core autophagy machinery. However, it is hard to tell the details about sequential recruitment of various components in the current assay as the assembly process happens very quickly.

i

Fig. 3i, *Atg2ab* DKO cells were starved for 2 h and cells were treated with 2% 1, 6-hexanediol for 20 min (H). Next the cells were recovered by washing with PBS and incubation with whole medium (HR) for 15min or 90 min. Then cells were stained with an antibody against p62 and ATG9A or p-ATG16L1. Scale bar, 10 μ m.

Supplementary Fig. 9 The recovery of p62 bodies after 1,6-hexanediol removal.

a, The number of indicated protein-positive puncta was quantified in Fig. 3i (3 independent experiments; 100 puncta were assessed per independent experiment).

b, *Atg2ab* DKO cells were starved for 2 h and cells were treated with 2% 1,6-hexanediol for 20 min (H). The cells were then recovered by washing with PBS and incubation with complete medium (HR) for 3min, 5min, 10min or 15min. Then cells were stained with an antibody against p62 and FIP200. Scale bar, 10 μ m.

3, How the p62 bodies specifically recruit these autophagy proteins? The authors also showed that p62 bodies recruit autophagy proteins like FIP200, ATG16L1, p-ATG16L1, ULK1, WIPI2, DFCP1 (Fig. 2a, b), while less or none colocalization was observed with ATG9A, ATG14 (Extended Data Fig. 2a, b). How this specificity is conferred? This can be discussed.

Response: Some of the autophagy proteins like FIP200 and ULK1 have direct interaction with p62^{3,4} while others like p-ATG16L1, WIPI2, DFCP1 may associate with p62 bodies through indirect interaction mediated by the contents in the condensates or vesicle recruitment around the droplet.

4, In Fig.1a, how the p62 droplet was isolated? The detailed information of this should be mentioned in methods section (centrifuge speed, time, washing?). Here, the droplet can be investigated by fluorescence to show that indeed the droplet contains p62 condensates/bodies.

Response: We have now added the detailed information of p62 droplet isolation in **Methods** section. Moreover, we added a fluorescence image to prove the presence of p62 in the droplet (**Supplementary Fig. 1a**).

Supplementary Fig. 1a p62 droplet formation by 3D fluorescence imaging. p62 droplet were labeled with wheat-germ agglutinin (WGA) -Alexa 488.

5, In Fig. 1b, the bands between 40-50kDa in Mix and FT group but not in Cytosol group seem interesting. Have the authors analyzed what are they?

Response: The band is the cleaved MBP tag (~42 kDa) derived from MBP-mCherry-p62 fusion protein.

6, Several autophagy proteins like ATG2A/B, ATG3, ATG13, ATG101 and ATG16L1 were found within p62 droplet in the mass spectrometry data. ATG16L1 is confirmed. Can the authors confirm one or two more proteins from ATG2A/B, ATG3, ATG13, ATG101? If appropriate antibodies are available.

Response: Following the reviewer's suggestion, we tested more ATG proteins for their colocalization with p62 droplet. As shown in **Supplementary Fig. 3b**, ATG2A/B, ATG3 and ATG13 displayed significant co-staining with p62 bodies, further supporting the high quality of the MS data.

Supplementary Fig. 3b EGFP-tagged ATG2A, ATG2B were transiently expressed in NRK cells. The cells were starved for 4 h, then stained with antibodies against GFP and p62 (upper panels). WT cells starved for 4 h were stained with antibodies against p62 and ATG3 or ATG13. Scale bar, 10 μ m.

7, The surrounding of autophagy proteins outside of p62 body (Fig. 2c) by ATG2KO is interesting. Can the authors block the condensates (for example, FIP200 and p62) ATG2KO cells, to see whether they are still phase separated condensates?

Response: We transiently expressed a p62 mutant (M404V; unable to bind ubiquitinated protein) that fails to form condensates in *Atg2ab* DKO cells and determined the distribution of ATG9A and p-ATG16L1 as well as FIP200 and WIPI2. As shown in **Supplementary Fig. 7**, we found that such mutant did not recruit ATG9A and p-ATG16L1 vesicles as well as FIP200 and WIPI2 puncta, suggesting that p62 body but not diffused form of p62 plays a critical role in the positioning and phase separation of the key indicated autophagy proteins. These results were further validated in p62 KO cells.

Supplementary Fig. 7 The recruitment of membrane vesicles and autophagy proteins by phase separation deficient p62-M404V mutant.

a-b, EGFP-tagged p62 or p62-M404V was transiently expressed in *Atg2ab* DKO cells (**a**) and p62 KO cells (**b**). The cells were then stained with antibodies against GFP, ATG9A, FIP200, p-ATG16L1 and WIPI2. Scale bar, 20 μ m.

8, In Extended Data Fig. 3a, cells were starved with DPBS (STA), and in Extended Data Fig. 3c, cells were starved with DMEM medium. It was planned for such difference in starvation conditions? The purpose of using STA should be mentioned.

Response: DPBS starvation is preferentially suitable for accelerating protein degradation by autophagy as revealed through Western blot analysis. In contrast, DMEM medium contains more nutrients and protein contents than DPBS. Thereby,

starvation by DMEM medium is milder and helps to maintain better cell morphology during imaging analysis.

9, In Fig4, the authors analyzed the proteins bound with the condensed liquid droplets. Can the authors also analyze the lipids bound with the condensed liquid droplets by Mass-spec, from WT cytosol or ATG2KO cells?

Response: Following the reviewer’s suggestion, we performed a lipidomic profiling of p62 droplet by mass spectrometry in both WT and *Atg2ab* DKO cells (**Supplementary Table 5**). The predominate lipids bound with p62 droplets were phosphatidylcholine (PC) and phosphatidylethanolamine (PE) (**Supplementary Fig. 12a, b**), which are known to be major membrane components of autophagosome^{5, 6}. Interestingly, the level of sphingomyelin (SM) of p62 body was dramatically increased in *Atg2ab* DKO cells (**Supplementary Fig. 12a, b**), consistent with the reports that sphingomyelin overload correlates with dysfunctional autophagosome formation^{7, 8}.

Supplementary Fig. 12 Lipidomic profiling of p62 bodies.

a-b, Composition of lipid classes that were considered for subsequent analysis in all of the samples detected by liquid chromatography–mass spectrometry/mass spectrometry.

10, What (1)-(3) in Fig.5a means? I guess these labeling was point to the quantification data in 5b. This should be mentioned in 5a legends.

Response: We apologize for not mentioning the labeling in the original Fig. 5a. Indeed, the numbers in the images correspond to the quantification data in Fig. 5b. We have mentioned this point in the revised Fig. 5b legend.

11, There several places in the text need to be confirmed or corrected. (a), “The class III phosphatidylinositol 3-kinase complex I (PI3KC3-C1) (primarily containing VPS34, ATG14 and Beclin1) is activated at phagophore membrane....”, here, should

VPS15 be included in PI3KC3-C1? (b), “...PI3KC3-C1 complex and ULK1 in the presence or absence of ATP and analyzed the condensate or supernatant fraction by immunoblotting assay (Fig. 5c, b).”, here, should “(Fig. 5c, b)” be “(Fig. 5c, d)”? (c), The title of Fig 5 legend, should “PI3K” be “PI3KC3-C1”?

Response: (a) We added VPS15 in PI3KC3-C1; (b) We corrected this typo as follows: (Fig. 5c, d); (c) It should be “PI3KC3-C1” and we corrected this typo in the revision.

References

1. Kageyama S, *et al.* p62/SQSTM1-droplet serves as a platform for autophagosome formation and anti-oxidative stress response. *Nat Commun* **12**, 16 (2021).
2. Komatsu M, *et al.* Homeostatic levels of p62 control cytoplasmic inclusion body formation in autophagy-deficient mice. *Cell* **131**, 1149-1163 (2007).
3. Turco E, *et al.* FIP200 Claw Domain Binding to p62 Promotes Autophagosome Formation at Ubiquitin Condensates. *Mol Cell* **74**, 330-346 e311 (2019).
4. Lim J, *et al.* Proteotoxic stress induces phosphorylation of p62/SQSTM1 by ULK1 to regulate selective autophagic clearance of protein aggregates. *PLoS Genet* **11**, e1004987 (2015).
5. Nakatogawa H. Mechanisms governing autophagosome biogenesis. *Nat Rev Mol Cell Biol* **21**, 439-458 (2020).
6. Andrejeva G, *et al.* De novo phosphatidylcholine synthesis is required for autophagosome membrane formation and maintenance during autophagy. *Autophagy* **16**, 1044-1060 (2020).
7. Gabande-Rodriguez E, Boya P, Labrador V, Dotti CG, Ledesma MD. High sphingomyelin levels induce lysosomal damage and autophagy dysfunction in Niemann Pick disease type A. *Cell Death Differ* **21**, 864-875 (2014).
8. Corcelle-Termeau E, *et al.* Excess sphingomyelin disturbs ATG9A trafficking and autophagosome closure. *Autophagy* **12**, 833-849 (2016).

Reviewer #1 (Remarks to the Author):

While the authors added new data and edited their manuscript, there is still a considerable number of points that definitively require their attention.

Regarding point #2: There is a mismatch of the number in the overlap between droplet and APEX: is this 157 (as stated in the rebuttal) or 158 (as stated on the figure)? Please double check and correct if necessary.

Regarding point #3: In Figure 1f, the authors need to provide information on how many proteins were actually found for each functional term (from the Metascale analysis). The same is true for Supplementary Figure 2f. Also, Supplementary Figure 2b is missing the molecular weight marker. What is the expression level of APEX2-GFP-p62 compared to endogenous p62. The readers need to know to which extent p62 is overexpressed.

Regarding point #3: For their GO analysis the authors should use a more conservative cutoff and only analyze the proteins in the overlap between the droplet and the APEX approach.

Regarding point #5: Based on the new Supplementary Figure 3a,b the majority of the tested proteins (RAB5C, STX17, STX6, STX12, RAB2A, STX3, ATG3) are clearly not recruited to p62 bodies. The lack of colocalization with p62 suggests that these candidates are all false positives. Were these potential false positives found in the overlap between the droplet and the APEX or just in the droplet approach? This poor validation rate is worrying for the robustness of this data set.

Regarding point #6: It is unfortunate that the authors do not present the fed and starved condition in one panel. In this split representation (Reviewer only Figure 1 and 2) it is impossible to judge the effect of starvation. Why are the insets shown with different magnifications?

Regarding point #8: While the authors now provide a rim measurement of p62 puncta, the quantification of how many p62 puncta show this enlarged halo is still missing. Also, not all autophagy markers behave the same way (e.g. for ATG14 and LC3 the p62 puncta just get bigger with no halo extending beyond the p62 core), yet the text does not discriminate between these clearly different phenotypes.

Regarding point #10: Without immune EM it is impossible to say whether the structure in DAB EM is actually a phagophore. This analysis needs to be repeated with the correct experimental setting. Importantly, the analysis in Supplementary Fig 8 needs to be done in p62 KO cells. Otherwise, endogenous p62 still contributes to the droplet formation. Moreover, the authors need to show that p62's LIR mediate the interaction with ATG16L1 and ATG9A. In addition, the authors need to control for the expression levels of wild-type and mutant (LIT and M404V) p62.

Regarding point #14: In Figure 4b, the authors absolutely need to control for the level of Cherry-p62 (wild-type and mutant) across the different fractions. If the mutant is less expressed, this will affect the pulldown efficiency. As noted before, the presence of endogenous p62 is worrying in this assay. This experiment should be done in p62 knockout background.

Reviewer #2 (Remarks to the Author):

The authors have appropriately responded to my concerns.

Reviewer #3 (Remarks to the Author):

The authors have addressed all of my concerns. They have significantly improved the manuscript and provided more information.

Point-by-Point Response to the Reviewers' Comments

We would like again to thank all the reviewers for their positive and constructive comments on our 1st edition of revision. While Reviewer #2 and #3 fully acknowledged the improvement of our revised manuscript, here we continually performed new experiments and analyses in the 2nd edition of revision to address Reviewer #1's additional concerns.

The changes in the manuscript are marked **red**, and the point-by-point responses to the reviewers' comments are provided below.

COMMENTS

Reviewer #1 (Remarks to the Author):

While the authors added new data and edited their manuscript, there is still a considerable number of points that definitively require their attention.

Regarding point #2: There is a mismatch of the number in the overlap between droplet and APEX: is this 157 (as stated in the rebuttal) or 158 (as stated on the figure)? Please double check and correct if necessary.

Response: We apologize for the typo in the previous rebuttal. The number in the figure was correct. There were 158 proteins overlapped between the droplet and APEX results.

Regarding point #3: In Figure 1f, the authors need to provide information on how many proteins were actually found for each functional term (from the Metascale analysis). The same is true for Supplementary Figure 2f. Also, Supplementary Figure 2b is missing the molecular weight marker. What is the expression level of APEX2-GFP-p62 compared to endogenous p62. The readers need to know to which extent p62 is overexpressed.

Response: We appreciate the reviewer's advice.

(1) We have changed the presentation style for new **Fig.1f** and new **Supplementary Fig. 2g** (previous Fig. S2f) which now included both the number information of enriched proteins and *p-value*. Moreover, the protein list was also included in the **Supplementary Table 3**.

Fig. 1f, Metascape enrichment of proteins enriched in the droplet (n = 590). The top 20 enriched main cluster terms of biological pathways were selected and presented.

Supplementary Fig. 2g, Metascape enrichment of the proteins enriched in both the droplet and the APEX labeling group (n=158). The top 20 enriched main cluster terms of biological pathways were presented. The sub clusters of “Vesicle-mediated transport” and “Selective autophagy” were shown.

(2) We have added the molecular weight marker in new **Supplementary Fig. 2c**.

Supplementary Fig. 2c, Coomassie brilliant blue staining showing the SDS-PAGE separation of the control and label samples and digested by trypsin. Each sample was cut into one fraction and analyzed by LC-MS/MS.

(3) The expression level of APEX2-GFP-p62 compared to endogenous p62 was shown in new **Supplementary Fig. 2a** by Western blot analysis. APEX2-GFP-p62 exhibited a reasonably higher expression than endogenous p62.

Supplementary Fig. 2a, Western blot analysis of the endogenous p62 and transfected APEX2-GFP-p62. Three biological replicates were performed for control and label groups, respectively. The control group was treated in parallel but without H₂O₂ stimulus compared to label group. The asterisk might indicate unspecific signal.

Regarding point #3: For their GO analysis the authors should use a more conservative cutoff and only analyze the proteins in the overlap between the droplet and the APEX approach.

Response: We thank the reviewer for this suggestion. We now utilized a more conservative cutoff for the GO analysis and only analyzed the overlapped proteins between the droplet and APEX results (n=158) instead of the enriched proteins (n=241) from APEX dataset in the old version. As shown in new **Supplementary Fig. 2g**, the focused pathways of “*Vesicle-mediated transport*” and “*Selective autophagy*” were more obviously enriched with much smaller *p-value* in the current version than those in the previous analysis. In the original version, these two pathways were ranked as 14th and 17th, while in the revised version, they ranked even higher as the 8th and 13th enriched terms, respectively.

Supplementary Fig. 2g, Metascape enrichment of the proteins enriched in both the droplet and the APEX labeling group ($n=158$). The top 20 enriched main cluster terms of biological pathways were presented. The sub clusters of “Vesicle-mediated transport” and “Selective autophagy” were shown.

Regarding point #5: Based on the new Supplementary Figure 3a,b the majority of the tested proteins (RAB5C, STX17, STX6, STX12, RAB2A, STX3, ATG3) are clearly not recruited to p62 bodies. The lack of colocalization with p62 suggests that these candidates are all false positives. Were these potential false positive found in the overlap between the droplet and the APEX or just in the droplet approach? This poor validation rate is worrying for this robustness of this data set.

Response: We apologize for not clearly presenting the imaging data in previous revision due to inappropriate region selection and color intensity adjustment. We have now chosen better images to present the colocalization status. As shown in **new Supplementary Fig. 3a, b**, all of the tested proteins showed colocalization with p62 bodies, suggesting that they were true hits of proteomics data. Notably, as vesicle trafficking is highly dynamic and different nature of each protein, the percentage, extent and spatial pattern of colocalization varied between tested proteins. Among them, RAB1A was found in the overlap between the droplet and the APEX data, while RAB5B, RAB5C, RAB2A, and STX17 just showed up in the droplet approach. Moreover, when exogenous tdTomato-p62 was co-transfected, the colocalization of p62 with RAB2A, RAB5C and STX6 was dramatically increased (**Reviewer only Fig. 1**), further supporting the validity of our proteomics hits.

Supplementary Fig. 3 Validation of colocalization of membrane vesicle proteins and ATG proteins with p62 bodies.

a, EGFP-tagged RAB5B, RAB5C, RAB1A, RAB2A, STX17, STX6, STX12 and STX3 were transiently expressed in NRK cells, starved for 4h, and cells were fixed and stained with antibodies against GFP and p62. Scale bar, 10 μ m.

b, EGFP-tagged ATG2A, ATG2B were transiently expressed in NRK cells. The cells were starved for 4 h, then stained with antibodies against GFP and p62 (upper panels). WT cells starved for 4 h were stained with antibodies against p62 and ATG3 or ATG13. Scale bar, 10 μ m.

Reviewer only Fig. 1 EGFP-tagged RAB5C, RAB2A and STX6 were transiently co-expressed with tdTomato-p62 in NRK cells. Scale bar, 10 μ m.

Regarding point #6: It is unfortunate that the authors do not present the fed and starved condition in one panel. In this split representation (Reviewer only Figure 1 and 2) it is

impossible to judge the effect of starvation. Why are the insets shown with different magnifications?

Response: In the new **Reviewer only Fig. 2**, we presented the data under fed and starved conditions in one panel, and the same magnification and presentation style were applied for the inset panel. The accumulation of p62 bodies and their colocalization with autophagic proteins were more pronounced in starved condition.

Reviewer only Fig. 2. The colocalization status of autophagic proteins with p62 bodies. NRK cells were starved for 4 h, and then stained with antibodies against FIP200, p-ATG16L1, ATG16L1, WIPI2 and p62. Scale bar, 10 μ m.

Regarding point #8: While the authors now provide a rim measurement of p62 puncta, the quantification of how many p62 puncta show this enlarged halo is still missing. Also, not all autophagy markers behave the same way (e.g for ATG14 and LC3 the p62 puncta just gets bigger with no halo extending beyond the p62 core), yet the text does not discriminate between these clearly different phenotypes.

Response: We have now provided a more detailed quantification by showing the number of p62 puncta and observed cells (new **Supplementary Fig. 4e, f**). We appreciate the reviewer's suggestion to clearly describe different phenotypes of autophagy markers for their colocalization with p62 puncta. We revised the corresponding part of the main text as follows: "In WT cells, core autophagy proteins FIP200 and WIPI2 were interspersed inside p62 bodies; however, they were excluded from the core of p62 droplets and concentrated at the periphery of p62 bodies in cells."

Other autophagic proteins such as LC3 and ATG14 rather exhibited a more flattened colocalization pattern with p62 puncta (Fig. 2c, d; Supplementary Fig. 4e-h)".

Supplementary Fig. 4e, The percentage of abnormal p62 bodies in either WT and *Atg2ab* DKO cells. (3 independent experiments; ~100 cells were assessed from 3 independent experiments). *P* values were calculated using the two-tailed, unpaired t-test, ** $P < 0.01$ **** $P < 0.0001$.

f, The numbers of abnormal p62 bodies in either WT and *Atg2ab* DKO per cell were quantified. (50 individual cells were assessed). *P* values were calculated using the two-tailed, unpaired t-test, ** $P < 0.01$ **** $P < 0.0001$.

Regarding point #10: Without immune EM it is impossible to say whether the structure in DAB EM is actually a phagophore. This analysis needs to be repeated with the correct experimental setting. Importantly, the analysis in Supplementary Fig 8 needs to be done in p62 KO cells. Otherwise, endogenous p62 still contributes to the droplet formation. Moreover, the authors need to show that p62's LIR mediate the interaction with ATG16L1 and ATG9A. In addition, the authors need to control for the expression levels of wild-type and mutant (LIT and M404V) p62.

Response: In previous rebuttal, we employed DAB EM to show isolation membrane or autophagosome formation around p62 bodies. We do think the current evidences are enough for this point and immune EM is not necessary because: 1) DAB EM already specifically marked p62 puncta, it will be technically challenging to double stain both p62 and autophagosome markers using immune EM; 2) The crescent-shaped double membrane structure around p62 puncta manifested typical morphology of isolation membrane (new **Reviewer Only Fig. 3a**); 3) Results from immunofluorescence assay further supported the presence of isolation membrane around p62 bodies (new **Reviewer Only Fig. 3b**); 4) As a classic receptor for selective autophagic cargos, p62 has been well documented to initiate autophagosome formation.

Reviewer Only Fig. 3, Isolation membrane around p62 bodies in WT cells by trans electron microscopy and immunofluorescence assay.

a, TEM image showing the DAB staining pattern in WT cells transiently transfected with GFP-APEX2-p62. Scale bar, 500nm, 200nm.

b, EGFP-tagged DFCP1 and tdTomato-p62 were transiently expressed in NRK cells, and then imaged. EGFP-tagged p62 were transiently expressed in NRK cells, starved for 4h and stained with antibodies against GFP and WIPI2/FIP200. Scale bar, 10 μ m.

For **Supplementary Fig. 8**, the experiments were indeed performed in *p62* knockout cells but we did not mention it clearly in the figure legend. We thank the reviewer for pointing it out and have corrected this information in the new figure legend. To further show that p62 LIR mut can still mediate the interaction with ATG16L1 and ATG9A, we performed an immunoprecipitation assay using *p62* knockout NRK cells that overexpressed either GFP-p62 WT or GFP-p62 LIR mut. As shown in new

Supplementary Fig. 8c, p62 LIR mut could successfully pull down ATG16L1 and ATG9A. In addition, the expression levels of wild-type and mutant (LIR mut and M404V mut) p62 were controlled at similar range and analyzed by Western blot analysis (new **Supplementary Fig. 7c**).

Supplementary Fig. 8c, EGFP-tagged p62 or EGFP-tagged p62 LIR mut was transiently expressed in *p62* KO cells. Immunoprecipitation was performed with IgG or GFP antibodies and protein A/G magnet beads, and the interacting proteins were analyzed by immunoblotting with FIP200, ATG9A, GFP, ATG16L1, LC3 and GAPDH antibodies.

Supplementary Fig. 7c, Western blot analysis of transient expression level of EGFP-tagged p62 or p62 M404V or p62 LIR mut in *p62* KO cells with antibodies against p62 and tubulin.

Regarding point #14: In Figure 4b, the authors absolutely need to control for the level of Cherry-p62 (wild-type and mutant) across the different fractions. If the mutant is lesser expressed, this will affect the pulldown efficiency. As noted before, the presence of endogenous p62 is worrying in this assay. This experiment should be done in *p62* knockout background.

Response: We re-performed this experiment as the reviewer suggested by using cell fractions derived from *p62* knockout NRK cells to exclude the influence of endogenous p62. The level of Cherry-p62 (both wild-type and Δ PB1 mutant) was also blotted. As shown in new **Supplementary Fig. 10e, f**, the previous conclusion in **Fig. 4b** drawn from *Atg2a/b* DKO cells was not affected when similar level of p62 was controlled in

p62 knockout cells. FIP200, ATG16L1, ATG9A and LC3 could co-precipitate with wild-type *p62* bodies, but not with *p62*- Δ PB1 mutant which lacks self-oligomerization domain and fails to be phase separated.

Supplementary Fig. 10 The sedimentation analysis as Fig. 4a, b was performed using *p62* knockout cell cytosol.

e, The sedimentation and supernatant from a were separated by centrifugation and analysed by western blot using antibodies against FIP200, ATG9A, ATG16L1, LC3, mCherry and GAPDH. S: supernatant, P: pellet.

f, *p62* body and different protein-interacting vesicles from a were co-stained with antibodies against *p62*, FIP200, p-ATG16L1, ATG9A and LC3. Scale bar, 10 μ m.

Reviewer #2 (Remarks to the Author):

The authors have appropriately responded to my concerns.

Response: We thank the reviewer for acknowledging our revision.

Reviewer #3 (Remarks to the Author):

The authors have addressed all of my concerns. They have significantly improved the manuscript and provided more information.

Response: We thank the reviewer for acknowledging our revision.

Reviewer #1 (Remarks to the Author):

I am happy that the authors have made sufficient effort to address all my remaining comments. Only, the authors should include the "Reviewer only Figures" as Supplementary Figures in the manuscript. They significantly contributed to the improvement of the manuscript and provide more information.

Point-by-Point Response to the Reviewers' Comments

We thank the reviewers again for their constructive comments to improve our manuscript. In the newest revision, we have modified the manuscript accordingly. The changes in the manuscript are marked **red**, and the point-by-point responses to the reviewers' comments are provided below.

COMMENTS

Reviewer #1 (Remarks to the Author):

I am happy that the authors have made sufficient effort to address all my remaining comments. Only, the authors should include the "Reviewer only Figures" as Supplementary Figures in the manuscript. They significantly contributed to the improvement of the manuscript and provide more information.

Response: We thank the reviewer for appreciating our efforts on the revision. Following the reviewer's suggestion, we have now included the "Reviewer only Figures" as new **Supplementary Fig. 3c** and **Supplementary Fig. 4** in the manuscript.